# Localization and functions of native and eGFP-tagged capsid proteins in HIV-1 particles

Ashwanth C. Francis[ID][1,2]*, Anna Cereseto[3], Parmit K. Singh[4,5], Jiong Shi[6], Eric Poeschla[7], Alan N. Engelman[4,5], Christopher Aiken[6], Gregory B. Melikyan[ID][2,8]*

**1** Institute of Molecular Biophysics, Department of Biological Sciences, Florida State University, Tallahassee, Florida, United States of America, **2** Division of Infectious Diseases, Department of Pediatrics, Emory University School of Medicine, Atlanta, Georgia, United States of America, **3** Center for Integrative Biology (CIBIO), University of Trento, Trento, Italy, **4** Department of Cancer Immunology and Virology, Dana-Farber Cancer Institute, Boston, Massachusetts, United States of America, **5** Department of Medicine, Harvard Medical School, Boston, Massachusetts, United States of America, **6** Department of Pathology, Microbiology, and Immunology, Vanderbilt University Medical Center, Nashville, Tennessee, United States of America, **7** Division of Infectious Diseases, University of Colorado Denver, Denver, Colorado, United States of America, **8** Children's Healthcare of Atlanta, Atlanta, Georgia, United States of America

* acfrancis@fsu.edu (ACF); gmeliki@emory.edu (GBM)

**Data Availability Statement:** All relevant data are within the manuscript and its Supporting Information files.

## Abstract

In infectious HIV-1 particles, the capsid protein (CA) forms a cone-shaped shell called the capsid, which encases the viral ribonucleoprotein complex (vRNP). Following cellular entry, the capsid is disassembled through a poorly understood process referred to as uncoating, which is required to release the reverse transcribed HIV-1 genome for integration into host chromatin. Whereas single virus imaging using indirect CA labeling techniques suggested uncoating to occur in the cytoplasm or at the nuclear pore, a recent study using eGFP-tagged CA reported uncoating in the nucleus. To delineate the HIV-1 uncoating site, we investigated the mechanism of eGFP-tagged CA incorporation into capsids and the utility of this fluorescent marker for visualizing HIV-1 uncoating. We find that virion incorporated eGFP-tagged CA is effectively excluded from the capsid shell, and that a subset of the tagged CA is vRNP associated. These results thus imply that eGFP-tagged CA is not a direct marker for capsid uncoating. We further show that native CA co-immunoprecipitates with vRNP components, providing a basis for retention of eGFP-tagged and untagged CA by sub-viral complexes in the nucleus. Moreover, we find that functional viral replication complexes become accessible to integrase-interacting host factors at the nuclear pore, leading to inhibition of infection and demonstrating capsid permeabilization prior to nuclear import. Finally, we find that HIV-1 cores containing a mixture of wild-type and mutant CA interact differently with cytoplasmic versus nuclear pools of the CA-binding host cofactor CPSF6. Our results suggest that capsid remodeling (including a loss of capsid integrity) is the predominant pathway for HIV-1 nuclear entry and provide new insights into the mechanism of CA retention in the nucleus *via* interaction with vRNP components.

**Funding:** This work was supported by NIH grants R21 AI145541 (to A.C.F.), R01 AI129862 (to G.B. M.), U54 AI150472 (to G.B.M. and A.C.F.), P50 AI150481 (to A.N.E. and C.A.), and R01 AI052014 (to A.N.E.). The funders had no role in study design, data collection and analysis, decision to publish, or preparation of the manuscript.

**Competing interests:** The authors have declared that no competing interests exist.

## Author summary

The timing, location and mechanisms of HIV-1 capsid disassembly which is referred to as uncoating remains unclear. Direct labeling of HIV-1 capsids, by incorporating a few green fluorescent proteins (GFP) tagged capsid protein (CA) into virions allows to image the spatio-temporal loss of HIV-1 CA during virus infection. However, the localization and functions of a few virion incorporated eGFP-tagged CA proteins remain unclear, since <50% of virus packaged CA proteins participate to form the conical capsid shell that protects the HIV-1 genome. Here we developed several approaches to test the localization and function of eGFP-tagged CA proteins in virions. We found that eGFP-tagged CA proteins are excluded from the conical capsid shell and that a subset of these proteins is associated with the viral ribonucleoprotein complex (vRNPs), through direct interactions between CA and vRNP components. eGFP-tagged CA is retained in the nucleus by virtue of vRNP association and is unlikely to report on HIV-1 capsid disassembly. We also found that HIV-1 capsids become permeabilized and are remodeled during their transport into the nucleus. Our study provides new insights into the ability of CA to interact with vRNPs for its retention in the nucleus and highlights capsid remodeling as a preferred pathway for HIV-1 entry into the nucleus.

## Introduction

The HIV-1 ribonucleoprotein complex (vRNP), which is composed of the viral RNA genome (vRNA) bound to the nucleocapsid (NC) protein and enzymes, reverse transcriptase (RT) and integrase (IN), is enclosed in a protective capsid shell. The capsid shell is assembled from ~240 capsid protein (CA) hexamers in a lattice-like arrangement that is interspersed with 12 CA pentamers to provide necessary curvature for a closed, cone-shaped structure (reviewed in [1,2]). We will refer to the capsid protein as CA, the assembled CA lattice shell as capsid, and the capsid together with its vRNP content as the viral core. Notably, of the ~3000 CA proteins packaged into virions in the form of Gag and Gag/Pol polyproteins, only ~1600 participate in the formation of the conical core following their proteolytic cleavage and virus maturation [2,3]. The localization and role of the remaining virion-incorporated CA molecules remains unknown.

Following HIV-1 entry into cells, the vRNA genome is reverse transcribed into linear double-stranded viral DNA (vDNA) by RT, which occurs within the confines of a reverse transcription complex (RTC) [4,5]. The RTC is then transported into the nucleus *via* a poorly understood nuclear import mechanism, where reverse transcription is completed [6–10]. The vDNA extremities are processed by IN to form the pre-integration complex (PIC) [11,12]. HIV-1 IN functions within the context of the intasome nucleoprotein complex [13,14] to catalyze vDNA integration into host chromatin. Here, we will collectively refer to cell-associated vRNPs, RTCs, and PICs, which are indistinguishable from one another by our imaging assays [10], as viral replication complexes (VRCs).

Critical interfaces formed between the capsid subunits are essential for key steps of HIV-1 infection, namely: (1) reverse transcription, (2) nuclear import, and (3) intranuclear transport to nuclear speckle compartments for integration into actively transcribing genes ([10] and reviewed in [1,15,16]). However, at some point after cellular entry, the capsid shell must dissociate to release the intasome for HIV-1 integration. The act of capsid disassembly has been traditionally referred to as uncoating, and the process of HIV-1 uncoating remains poorly understood. For example, it is unclear where in the cell productive uncoating occurs

(cytoplasm, during nuclear entry, post nuclear import, or some combination thereof) and whether CA is progressively lost from the capsid shell or if disassembly occurs in a synchronous fashion (reviewed in [1]). Moreover, to account for the role of nuclear CA in targeting VRCs to speckle-associated domains (SPADs) for integration [10], recent models have proposed 'capsid remodeling' in lieu of uncoating [17,18], whereby the conical capsid changes its composition, shape and/or structure upon nuclear import.

Importantly, regardless of where uncoating is initiated and the extent to which capsid is modified during nuclear import, at least a subset of hexameric CA lattice should be retained by nuclear VRCs for its transport inside the nucleus. HIV-1 intranuclear transport is determined, in large part, by interactions between the host cleavage and polyadenylation specificity factor 6 (CPSF6) (reviewed in [16]) and a binding pocket formed by adjoining CA molecules within the hexameric ring [19,20], which we refer to as the intra-hexamer pocket. However, it is currently unclear if intact capsids are required, or if partial CA hexameric lattices (remnants of capsid remodeling) would suffice for CPSF6 interactions in the nucleus. In the latter scenario, a mechanism of how a partial CA lattice is retained by nuclear VRCs remains unknown, and a direct interaction between CA and VRC components (RT, IN, NC and viral RNA/DNA) has not been demonstrated.

Recent cryo-electron tomography (cryo-ET) studies of intact nuclear pore complexes (NPCs) revealed that the inner diameter of these pores is variable and can reach ~ 64 nm [21–23]–a size that could, in principle, permit the translocation of intact viral cores (~60x30x90 nm, [2]). Indeed, recent correlative light-electron microscopy (CLEM) studies found evidence for cone-shaped HIV-1 cores docked at NPCs. In these studies, the majority of nuclear HIV-1 complexes were deformed/broken, with a few apparently intact cone-shaped cores with associated interior electron-density possibly corresponding to VRCs [22,24]. However, the relevance of these nuclear cone-shaped cores to productive HIV-1 infection remains unclear.

Various HIV-1 CA labeling approaches have been developed to examine uncoating in the context of productive infection. Live-cell imaging has been leveraged to visualize loss of fluorescent CA markers from single virus particles that establish infection [6,7,25–28]. However, discrepant findings related to HIV-1 uncoating have been reported. Functionally relevant HIV-1 uncoating has been proposed to occur in the cytoplasm, shortly after viral-cell fusion [26], at the nuclear pore, prior to nuclear entry [6,27], and in the nucleus [7,25]. These conflicting findings largely stem from the challenges associated with non-invasive fluorescent labeling of HIV-1 CA and from the interpretation of the apparent loss of CA markers in living cells.

Recently, N- and C- terminal eGFP fusion proteins with CA were constructed and used to assess HIV-1 uncoating in cells [7,29]. However, CA tagging with eGFP severely impaired HIV-1 infectivity, necessitating the use of mixed cores containing a >10-fold excess of untagged CA to generate infectious virions [7,29]. Imaging revealed that a major fraction of virion-incorporated eGFP-CA signal was released after virus membrane permeabilization, leaving a small subset of eGFP-tagged CA associated with viral cores. In live/fixed cell experiments this small core-associated fraction of tagged CA was retained after nuclear import. While Zurnic *et al.* concluded that C-terminal eGFP-tagged CA in the nucleus was not functionally relevant for integration [29], Burdick *et al.* observed disappearance of single N-terminally eGFP-tagged CA near the sites of integration and concluded that intact or nearly intact cores were transported into the nucleus, for nuclear uncoating [7].

Here, we investigated the mechanism of eGFP-tagged CA incorporation into the HIV-1 capsid. Importantly, we find that eGFP-tagged CA is effectively excluded from the CA lattice and that a subset of this fusion protein is co-packaged with vRNPs inside the capsid shell. This finding argues against using eGFP-tagged CA as a reliable marker for functional HIV-1 uncoating. By co-immunoprecipitation experiments we show that native CA can interact with

vRNP components, suggesting a mechanism for the incorporation, and for the retention of native and eGFP-tagged CA in nuclear VRCs. Failure of eGFP-tagged CA to report uncoating prompted additional investigation into the integrity of HIV-1 capsids during nuclear import. Consistent with previous reports [30,31], our results indicate that HIV-1 IN, which is a part of the vRNP [32], becomes accessible to host factors at the nuclear pore. Furthermore, we find that HIV-1 capsids containing a mixture of mutant (N74D) and wild-type (WT) CA interact with the CPSF6-358 restriction factor in the cytoplasm but fail to efficiently engage full-length CPSF6 in the nucleus, demonstrating differences in HIV-1 capsids in the cytoplasm versus nucleus. Collectively, our results highlight capsid remodeling as a predominant pathway for HIV-1 nuclear import and provide new insights into the previously unappreciated CA-vRNP interactions, suggesting a basis for retaining CA in the nucleus.

## Results

### Development and validation of CAeGFP fusion protein to fluorescently label HIV-1 capsid

We constructed a plasmid encoding HIV-1 MA-CAeGFP fusion protein that is incorporated into virions through MA-driven co-assembly with unlabeled Gag/GagPol polyproteins encoded by the viral backbone (Fig 1A). The native protease cleavage site between MA and CAeGFP allows the release of fluorescent CAeGFP upon proteolytic cleavage during virus maturation. MA-CAeGFP lacks the nucleocapsid (NC) and p6 domains of Gag. Therefore, co-incorporation of a second fluorescent marker, Vpr-IN fused to super-folder-mCherry (INsfCherry), which is incorporated *via* Vpr interactions with the p6 domain of unlabeled Gag/GagPol [33], was used to identify virions containing a mixture of unlabeled CA and CAeGFP. This labeling approach differs from the recently described N-terminal eGFP-CA fusion protein (Fig 1B) (hereafter referred to as HGFP-CA, [7]), and C-terminal CAeGFP [29] (S1A Fig), which are expressed in the context of full-length Gag/GagPol.

In this study, we compared N-terminally labeled HGFP-CA and C-terminally labeled CAeGFP approaches to label HIV-1 capsids. We incorporated HGFP-CA or CAeGFP into virions containing a >10-fold excess of unlabeled Gag/GagPol derived from an HIV-1 back-bone and analyzed the effects of virus labeling on maturation and infectivity. Analysis of fluo-rescently tagged intact viral particles bound on poly-L-lysine treated glass revealed that both CAeGFP and HGFP-CA were efficiently incorporated into HIV-1, as evidenced by >89% colocalization with the vRNP-marker INsfCherry (Fig 1C). Immunoblot analysis revealed that the incorporated fluorescent MA-CAeGFP and HGFP-CA precursors were processed similarly by the viral protease producing GFP-tagged CA proteins, and that their incorporation did not noticeably affect the overall efficiency of Gag/GagPol processing (Fig 1D and 1E). Further-more, consistent with the recent reports [7,29], only a modest reduction (<2-fold) in single-round infectivity was observed for fluorescent CA-labeled viruses compared to unlabeled viruses (Fig 1F). Thus, neither HGFP-CA nor CAeGFP significantly perturb virus infectivity when co-incorporated with an excess of unlabeled Gag/GagPol.

### eGFP-tagged CA remains associated with HIV-1 replication complexes in the nucleus

Next, we asked whether virus-incorporated CAeGFP and HGFP-CA associate with the viral core by comparing the eGFP-signal of immobilized viral particles to those observed after per-meabilization of the viral membrane with saponin [34]. Saponin-formed pores expose the virus lumen and release a soluble CA pool that is loosely trapped within the viral envelope.

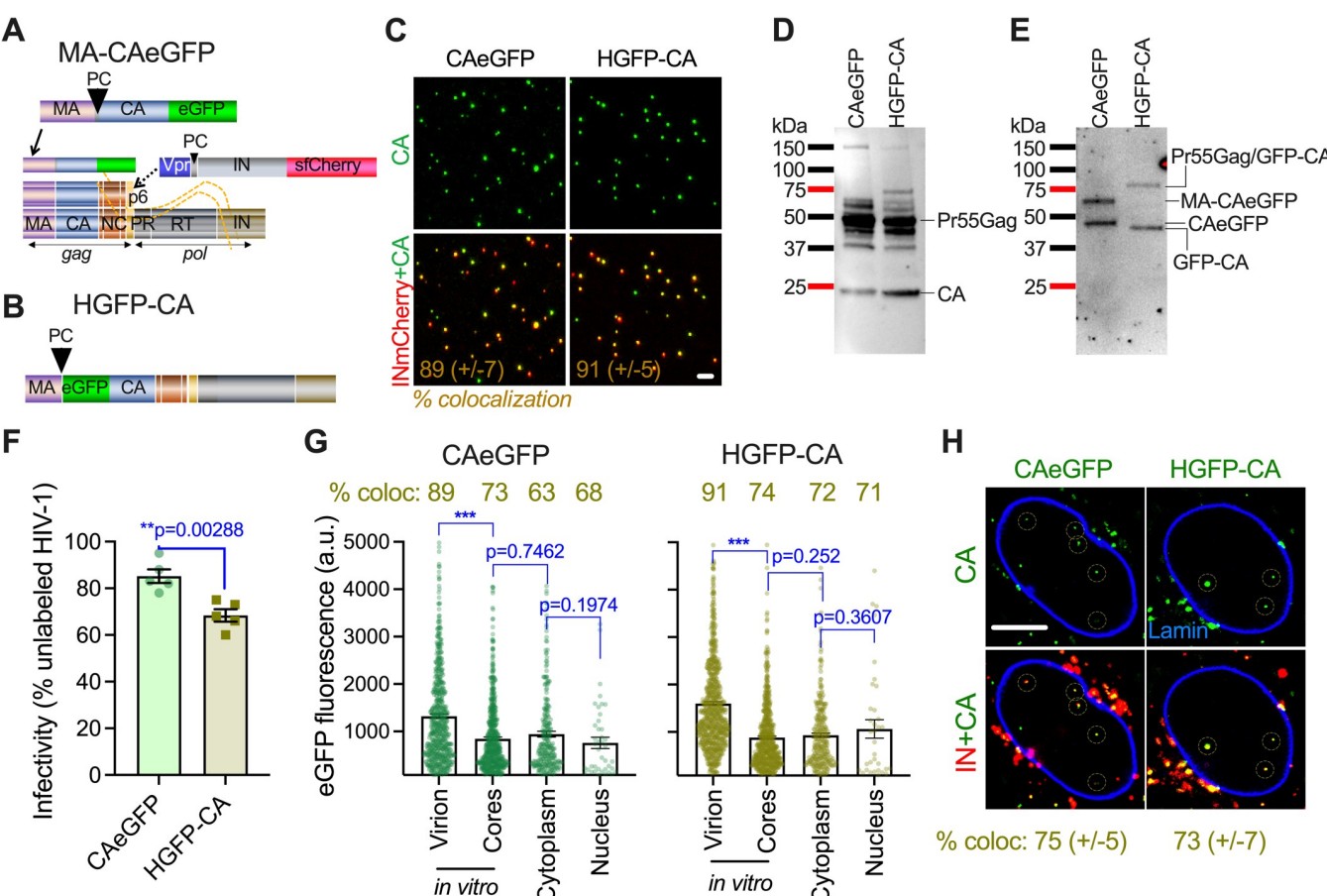

**Fig 1. N- or C-terminally GFP labeled CA proteins are efficiently processed in virions and remain associated with HIV-1 replication complexes in the nucleus.** (A) Schematic of C-terminal tagging of CA through a MA-CAeGFP fusion construct. MA drives co-assembly and virion incorporation of MA-CAeGFP with unlabeled Gag/GagPol encoded by the HIV-1 backbone. The protease cleavage site (PC) between MA and CA releases mature fluorescent CAeGFP upon virus maturation. The Vpr-INsfCherry fusion construct supplied *in trans* is incorporated *via* interaction with the p6 domain of unlabeled Gag. The resulting viral particles co-labeled with CAeGFP and INsfCherry contain a mixture of native and eGFP-labeled CA proteins. (B) Schematic of CA tagging strategy reported by Burdick et al., PNAS 2020 based on the N-terminal GFP-CA fusion protein expressed in the context of full-length Gag/GagPol. (C) Efficiency of HIV-1 co-labelling by GFP-tagged CA and INmCherry. The % colocalization of CAeGFP with INmCherry (+/- SD from 5 experiments) is shown in gold. Scale bar is 2 μm. (D, E) Western blot of viruses labeled with CAeGFP or HGFP-CA showing efficiently processed Gag/GagPol probed with anti-HIV IgG (D) and release of GFP-tagged CA fusion proteins probed with anti-GFP antibodies (E). (F) Specific infectivity of HIV-1 virions labeled with the different GFP-tagged CA constructs was determined in TZM-bl cells. Viruses were normalized for RT activity and infectivity was determined by luciferase assays in triplicate at 48 hpi. The % infectivity of HGFP-CA and CAeGFP labeled viruses was determined by setting luciferase relative light units (RLUs) of unlabeled HIV-1 to 100% for each experiment. The mean infectivity and SEM from 5 experiments is shown. (G) Fluorescence intensity distribution of GFP-tagged CA proteins associated with INmCherry puncta in intact virions, after virus membrane permeabilization with saponin (cores) and at 4 hpi in the cytoplasm and nucleus. The average object-based colocalization of INsfCherry and GFP-tagged CA from 5 independent experiments is shown above each condition. (H) Single (*top panel*) and merged (bottom panel) images (1 μm projection of the central plane) showing the distribution of GFP-tagged CA (green) and colocalized INsfCherry (red) labeled HIV-1 VRCs at 12 hpi in the nucleus of TZM-bl cells that stably express SNAP-Lamin (blue). Cells were infected at MOI 0.5. Quantitation of eGFP-intensities in nuclear VRCs between 4 (G) and 12 h (H) are shown in related S1B Fig. Scale bar is 5 μm. Statistics: In (F), data were analyzed using student t-test; in (G), a non-parametric Mann-Whitney Rank Sum test was used. p-values: ** <0.01 and *** <0.001.

The majority of virion-incorporated eGFP signal was released after detergent treatment, leaving <25% of the total eGFP-signal retained by cores (Fig 1G). These results are consistent with prior reports [7,29] that most of virus-incorporated eGFP-tagged CA is not associated with the core. From the 1:10 ratio of tagged-to-untagged CA expression constructs and loss of >75% of eGFP signal following saponin treatment, we estimate that core-associated eGFP-tagged CA accounts for about <2.5% of total CA packaged into virions.

Notably, >70% of INsfCherry2 labeled HIV-1 cores retained a fraction of eGFP signal after saponin treatment (Fig 1G). We sought to determine if this core-associated eGFP-tagged CA is retained following the entry of HIV-1 core into the cytoplasm and in the nuclei of infected cells, as has been previously reported [7,29]. TZM-bl cells infected with HIV-1 pseudoviruses co-labeled with INsfCherry and CAeGFP or HGFP-CA markers were imaged at 4 hours post-infection (hpi), and the retention of eGFP-CA signal in IN-labeled VRCs residing in the cytoplasm or in the nucleus was analyzed. In agreement with the recent reports [7,29], a subset of eGFP-tagged CA remained co-localized with >60% of cytoplasmic and nuclear IN-labeled VRCs by 4 hpi, a time point when approximately half of VRCs are imported into the nucleus [6,27]. Notably, the mean eGFP intensities detected in the nucleus were similar to intensities of single viral cores in the cytoplasm and *in vitro*, after viral membrane permeabilization (Fig 1G). Thus, core-associated eGFP-tagged CA is retained in VRCs after their import into the nucleus.

Importantly, the colocalization and mean eGFP-tagged CA intensities remained unchanged when nuclear IN-labeled VRCs were imaged at a later time-point (12 hpi, Figs 1H and S1B). Because reverse transcription peaks by 8 to 10 hpi in TZM-bl cells under these infection conditions [6,27], it appears that eGFP-tagged CA remains associated with the nuclear VRCs, even after completion of reverse transcription, which is generally thought to destabilize the HIV-1 core [26,34–36].

## eGFP-tagged CA does not co-assemble with unlabeled CA into a capsid lattice

To test whether eGFP-tagged CA incorporates into the capsid shell, we probed its accessibility to host factors that recognize the capsid lattice. The cytosolic protein cyclophilin A (CypA) interacts with the CypA-binding loop of CA exposed on the surface of the HIV-1 core (reviewed in [1]). Other cellular proteins, including CPSF6, SEC24C, and Nup153, interact with the common intra-hexamer CA binding pocket on capsid [19,37,38]. Accordingly, the G89V mutation in the CypA-binding loop of CA and the N74D mutation within the intra-hexamer pocket render HIV-1 insensitive to restriction by the CypA domain-containing TRIM-Cyp [39,40] and by cytoplasmic CPSF6-358 (a truncated form of the nuclear CPSF6 protein) [41,42], respectively.

To test if eGFP-tagged CA proteins can interact with TRIMCyp or CPSF6-358 in the cytoplasm, we incorporated eGFP-tagged WT CA proteins into resistant viruses carrying, respectively, the G89V or N74D mutant CA proteins. We reasoned that, if eGFP-tagged CA co-assembles with the capsid lattice, it would be exposed to the restriction factors, and thus render the G89V and N74D mutant viruses sensitive to restriction by TRIMCyp and CPSF6-358, respectively (illustrated in Figs 2A and S2B). As a control, we incorporated unlabeled WT CA by co-transfecting virus-producing cells with mutant and WT CA-encoding psPAX2 plasmids at the ratio of 10:1. Western blot analysis revealed that the WT CAeGFP precursor was proteolytically processed to release CAeGFP after its incorporation into N74D and G89V CA mutant viruses (S2A Fig).

Target cells constitutively expressing the truncated cytosolic CPSF6-358 fragment [41] or control cells were infected with HIV-1 pseudoviruses produced with 100% WT or N74D CA plasmid, or with 10% of either unlabeled WT CA or eGFP-tagged WT CA plasmids. A single round infectivity assay revealed that, as expected, viruses produced with 100% of WT CA were highly sensitive to CPSF6-358 restriction, while those containing only N74D CA were completely resistant to restriction (Fig 2B). Similar to the pure WT CA viruses, the mixed WT/ N74D CA viruses produced with 10% unlabeled WT CA were sensitive to CPSF6-358

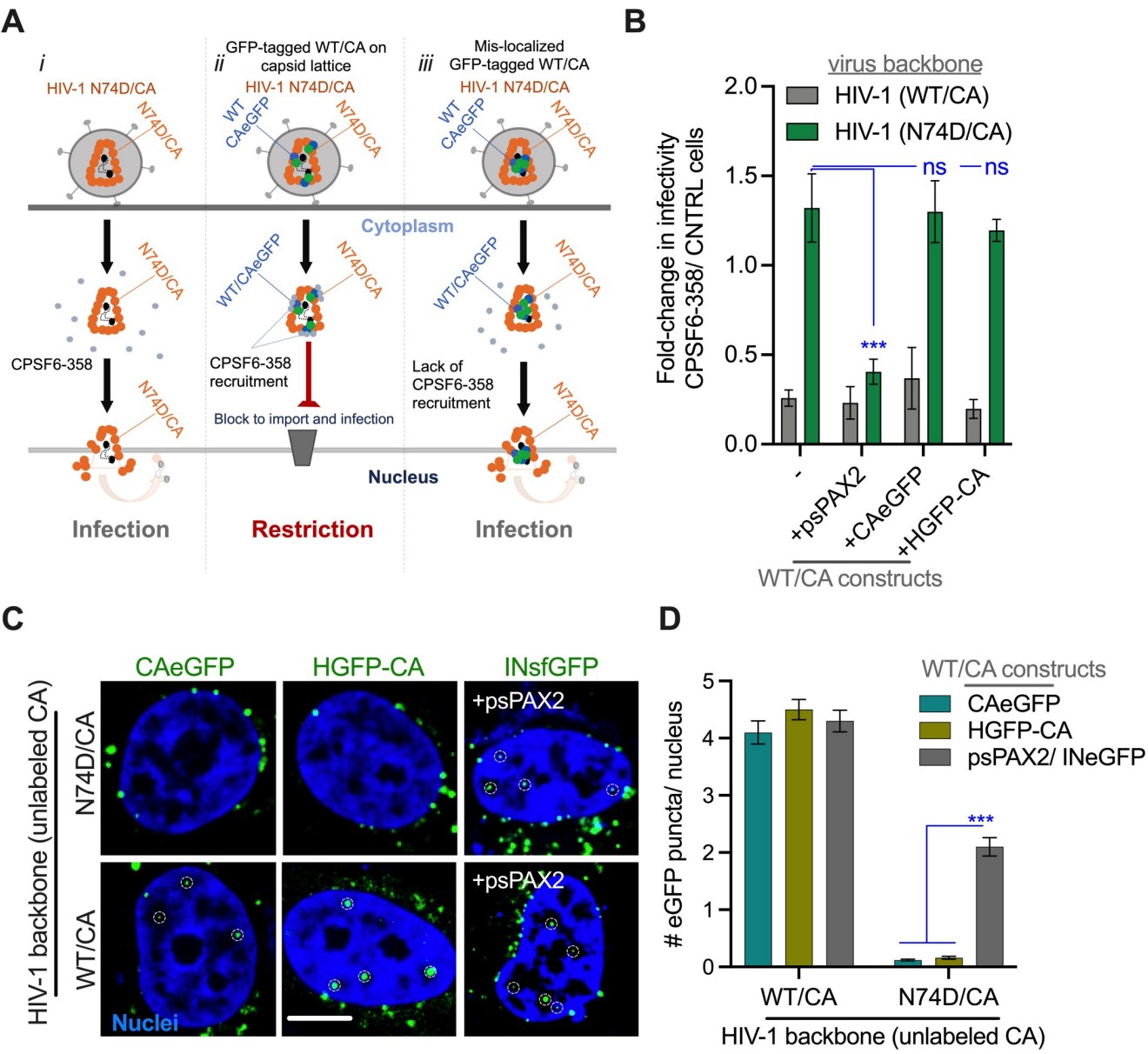

**Fig 2. GFP-tagged CA is not accessible on the capsid lattice for host-factor interactions. (A)** Strategy to test the localization of viral core-associated GFP-tagged CA. The ability of GFP-tagged CA to render N74D CA virions sensitive to cytosolic CPSF6-358 in TZM-bl cells. Three possible scenarios are illustrated: *(i)* In the absence of WT CA (tagged or untagged), N74D CA does not bind to CPSF6-358 and therefore resists restriction. *(ii)* If GFP-tagged WT CA incorporates into the capsid shell consisting of the N74D CA mutant, it is expected to interact with CPSF6-358 and restrict N74D virus infectivity. *(iii)* Mis-localization of GFP-tagged CA, e.g. with the vRNP inside the viral core, will not sensitize N74D virus to CPSF6-358. **(B)** Single-round infectivity data shows that GFP-tagged CA is inaccessible to CPSF6-358, while 10% unlabeled WT CA in N74D viruses efficiently interacts with CPSF6-358 resulting in restriction of N74D infectivity. As controls, HIV-1 with 100% unlabeled WT or mutant CA or a mixture of 10% unlabeled WT (psPax2) and 90% N74DCA were used. Infectivity is normalized to that in control cells that do not express CPSF6-358. The average values from 4 experiments and STD are shown. Single Z-slice images **(C)** and quantification **(D)** show the lack of nuclear penetration of WT or N74D CA VRCs labeled with N- (HGFP-CA) or C-terminally (CAeGFP) tagged WT CA. Note, N74D CA VRCs labeled with eGFP-tagged CA remain at the nuclear periphery (*top left- and middle panels*). Control infections with 10% of unlabeled CA (psPAX2) is able to mediate penetration of INsfGFP-labeled N74D VRCs (marked by white dashed circles, *top right panel*). The distribution of eGFP-tagged CA and INsfGFP-labeled VRCs of HIV-1 WT CA is shown for comparison (bottom panels). Scale bar in (C) is 5 μm. The distribution of fluorescent HIV-1 puncta in (D) is plotted as mean and SEM for >120 nuclei from 3 experiments. Statistics: in (B) student's t-test was used; data in (D) were analyzed using a non-parametric Mann-Whitney Rank Sum test.

restriction, consistent with their co-assembly into the CA lattice (Fig 2B, +psPAX2). These data indicate that phenotypically mixed CA lattices containing as little as 10% WT CA are sufficient for near-full restriction by cytoplasmic CPSF6-358. Strikingly, incorporation of eGFP-tagged CA proteins into N74D viruses did not alter their sensitivity to CPSF6-358 (Fig 2B). These results imply that, unlike the unlabeled CA, eGFP-tagged CA does not co-assemble with the N74D CA lattice and thus does not interact with cytoplasmic CPSF6-358.

Similarly, we found that the CypA-binding loop of eGFP-tagged CA is not exposed on the surface of post-fusion cores, as evidenced by their inability to recruit TRIMCyp. Consistent with the previous findings [43], viruses produced with 100% G89V CA were completely resistant to TRIMCyp restriction, whereas mixed WT/G89V CA viruses with 10% unlabeled WT CA (S2C Fig, +psPAX2) and 100% WT CA containing viruses (S2C Fig, (-)) were nearly equally sensitive to this restriction factor. In contrast, eGFP-tagged WT CA in the context of phenotypically mixed virions did not interact with TRIMCyp, as evidenced by the lack of effect on G89V infectivity in TRIMCyp expressing cells (S2C Fig).

We further probed the ability of N74D virus-incorporated eGFP-tagged or unlabeled WT CA to engage endogenous full-length CPSF6 in the nucleus, which is required for VRC transport into the nucleoplasm, away from the nuclear envelope [10,27,44]. Toward this goal, we asked if the mixed WT/N74D CA viruses produced with 10% of unlabeled WT CA or eGFP-tagged WT CA were transported at least 0.5 µm away from the nuclear envelope. As expected, in control infections with 100% WT CA, the eGFP-tagged CA or INsfGFP labeled VRCs localized to the nuclear interior at 4 hpi (Fig 2C). The incorporation of 10% unlabeled WT CA was sufficient to rescue the N74D CA mutant interactions with CPSF6, resulting in measurable penetration of a subset of INsfGFP-labeled VRCs into the nucleus, although this penetration was less apparent than that of pure WT CA viruses (Fig 2C and 2D). In contrast, but consistent with our results from the cytoplasmic restriction assay (Fig 2B), eGFP-tagged WT CA failed to interact with endogenous full-length CPSF6, as evidenced by the unchanged peripheral localization of N74D complexes (Fig 2C and 2D). Collectively, these results show that core-associated eGFP-tagged CA is not accessible to critical host factors in the cytoplasm and the nucleus of target cells. The eGFP-tagged CA is thus unlikely to form hexameric assemblies as a part of the capsid shell.

## eGFP-tagged CA disappears along with IN-labeled nuclear VRCs at the time of integration

Our finding that eGFP-tagged CA is excluded from the capsid shell prompted further investigation of the utility of eGFP-tagged CA as a marker of nuclear uncoating. We have previously shown that disappearance of IN-labeled nuclear VRCs correlates with HIV-1 integration and subsequent expression of eGFP from integrated vDNA [6,27]. The correlation of IN-labeled VRC signal disappearance with integration was also reported by others [45,46]. We performed live-cell imaging of HIV-1 pseudoviruses co-labeled with eGFP-tagged CA and INsfCherry, and analyzed infected cells for the disappearance of the eGFP-labeled CA and INsfCherry signals.

Single virus tracking revealed that eGFP-tagged CA was retained upon nuclear import and remained colocalized with INsfCherry labeled VRCs throughout nuclear trafficking (S3A–S3D Fig). Following extended nuclear co-trafficking of co-labeled CA puncta, simultaneous loss of both signals was observed for all CAeGFP (n = 24) and HGFP-CA (n = 18) VRCs (S3E Fig). A total of 26 and 20 cells became infected with CAeGFP and HGFP-CA labeled viruses, respectively. Moreover, the disappearance of nuclear eGFP-tagged CA/INsfCherry complexes, which was observed in over 22% of cells in control infections, was largely reduced (<5%) upon

inhibition of integration with 10 μM raltegravir (S3F Fig, +RAL). A strong correlation between nuclear VRC disappearance and eGFP reporter expression in control cells contrasts with the lack of disappearance in RAL treated cells. These results suggest that eGFP-tagged CA and IN fluorescent signals disappear near simultaneously, likely at the time of functional integration.

## eGFP-tagged CA associates with vRNPs inside viral cores

To probe the possibility of eGFP-tagged CA association with vRNPs, we employed an *in vitro* uncoating assay [34]. This assay relies on CypA-DsRed (CDR) protein that is incorporated into pseudoviral particles through avid CA binding, which allows for visualization of the loss of CA from HIV-1 cores in real time. Here, we co-incorporated eGFP-tagged CA and CDR into viruses containing WT CA or the unstable K203A CA mutant [34,47]. The CDR and eGFP signals were analyzed for coverslip-immobilized virions, following viral membrane permeabilization with saponin (+SAP, Fig 3A). We have previously observed that, in contrast to WT CA cores which disassemble (lose CDR and CA) with a half-time of 12 min, K203A mutant cores disassembled immediately upon saponin or detergent treatment *in vitro*, or shortly after virus-cell fusion in live cells [34]. These mutant cores thus allow detection of fluorescent markers that remain associated with vRNPs after uncoating.

Similar to the results shown in (Fig 1G), virus membrane permeabilization released the majority (~75%) of eGFP-tagged CA (Fig 3B and 3D). Importantly, residual eGFP signal was detectable in the unstable K203A mutant particles, implying that eGFP-tagged CA remains bound to K203A vRNPs after uncoating *in vitro*. Moreover, the detection of approximately equal residual eGFP signal in permeabilized WT and K203A virions is consistent with the notion that nearly all remaining GFP-tagged CA is vRNP-associated and not incorporated into the CA lattice.

In contrast to the vRNP-associated eGFP-tagged CA, saponin treatment caused a dramatic loss of CDR from unstable K203A cores, while only modestly decreasing the CDR signal from WT cores (Fig 3B, 3C and 3E). A stark difference between the loss of eGFP and CDR from WT virions upon permeabilization suggests that, intriguingly, CDR does not associate with the CA or eGFP-tagged CA pool that is released upon saponin treatment and is therefore almost exclusively associated with the assembled CA lattice. This result, combined with the nearly complete loss of CDR from K203A vRNPs (Fig 3E), indicates that CDR does not tightly associate with eGFP-tagged CA in vRNPs. Therefore, avid-interactions between CDR and CA appear to be exclusive to the assembled capsid lattice and, unlike the residual eGFP-tagged CA, CDR is released upon capsid uncoating *in vitro*.

We next asked whether CDR is recruited to vRNPs by eGFP-tagged CA (Fig 3A). Permeabilized viruses were treated with cyclosporin A (CsA), which binds to CypA with sub-nanomolar affinity and displaces it from intact cores [34]. We reasoned that a subset of CDR recruited by eGFP-tagged CA to the vRNP inside intact cores should be protected from CsA and thus would not be displaced by this treatment. Accordingly, CsA treatment of permeabilized viruses (+SAP+CsA) caused incomplete (~2.5-fold reduction) release of CDR from WT cores (Fig 3E), without noticeably affecting the eGFP signal (Fig 3D). This result confirms that, when combined with eGFP-tagged CA, a fraction of CDR is protected from CsA and is thus likely located inside intact capsids composed of WT CA. Also, CsA treatment of unstable K203A complexes that uncoated immediately after saponin treatment resulted in a modest but significant reduction of the already very low CDR signal down to background level (Fig 3E).

To further test if a CsA-resistant CDR pool is, in fact, vRNP-associated, we made a Vpr-CL-INmNeonGreen-CA (Vpr-CL-INmNG-CA) fusion protein (S4A Fig). This protein is incorporated into virions *via* Vpr and, upon virus maturation, is cleaved by the HIV-1

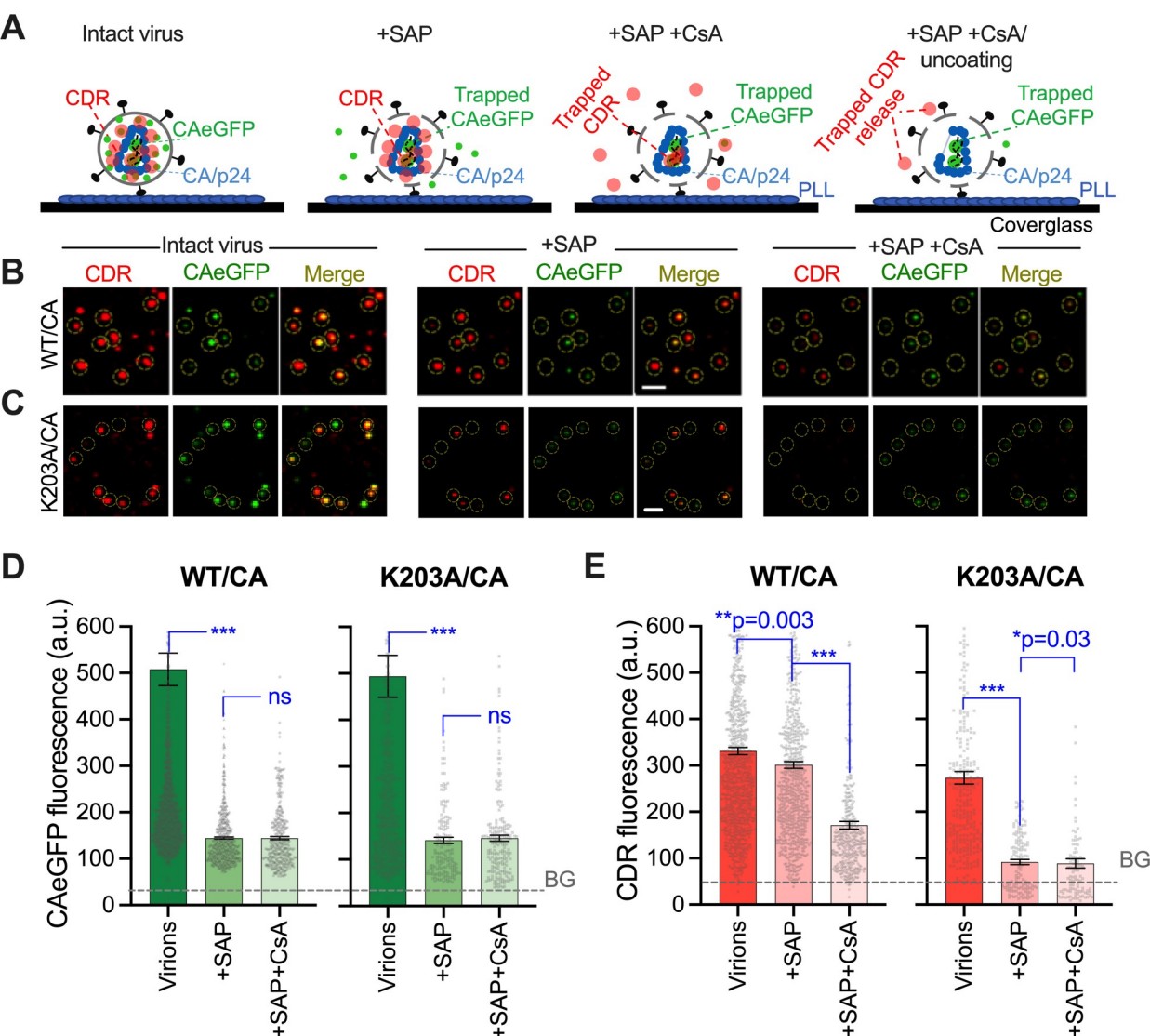

**Fig 3. A subset of GFP-tagged CA localizes inside the conical capsid core.** **(A)** A strategy to test localization of GFP-tagged CA in single virions. HIV-1 co-labeled with CAeGFP and CypA-DsRed (CDR) are bound to a poly-lysine coated glass (left). Following membrane permeabilization with saponin (SAP; 100 µg/ml) the major fraction of un-polymerized fluid-phase CAeGFP is released, while CDR remains associated with cores through high avidity binding (second from left). Addition of CsA (5 µM) displaces surface-accessible CDR from cores, whereas CDR trapped inside intact cores by virtue of association with CAeGFP in vRNPs is unaffected (second from right) and is displaced only after loss of core integrity or after the initiation of uncoating (right). **(B, C)** Images of co-labeled WT CA (B) or K203A CA (C) containing HIV-1 before (left) and after treatment with SAP (middle) or SAP + CsA (right). Scale bars in (B) and (C) are 2 µm. **(D, E)** Analysis of CAeGFP (D) and CDR (E) fluorescence in single virions, after virus membrane permeabilization (+SAP) and after CsA (+SAP+CsA) treatment. The background in eGFP (50 a.u.) and CDR (80 a.u.) channels are marked by dashed blue lines. Data in (D and E) are mean and SEM of the cumulative distribution of fluorescent intensities from 4 experiments. Statistics in (D and E) non-parametric Mann-Whitney Rank Sum test, ns p>0.05; * p<0.05; ** p<0.01 and *** p<0.001.

protease at the engineered protease cleavage site (CL), producing a chimeric INmNG-CA protein (S4B Fig). The INmNG-CA is expected to preferentially package into vRNPs through association with the viral genome and oligomerization with unlabeled IN and, because of this, to recruit CDR to vRNPs [10,27,48]. We found that INmNG-CA incorporation did not affect HIV-1 infectivity beyond the observed <2-fold reduction seen for the control INmNG labeled viruses (S4C Fig).

We selectively localized CDR to vRNPs (and not to the capsid shell) by co-incorporating CDR through the INmNG-CA WT into a G89V CA virus (which by itself is incapable of binding CDR (S4D Fig) [34]), and asked if this vRNP-associated CDR is resistant to saponin and CsA treatment. As controls, CDR was co-incorporated into G89V viruses with Vpr-CL-INmNG, that lack CA fusion protein. As expected, the control G89V viruses labeled with INmNG did not incorporate CDR, whereas the INmNG-CA chimera efficiently recruited CDR into G89V viruses resulting in >80% co-labeling (S4E Fig). Furthermore, CDR incorporated through INmNG-CA into G89V CA viruses was resistant to both saponin and CsA treatment (S4F Fig). These results confirm that CA, and therefore CDR, can be directed to the vRNP when linked to IN and that this CDR pool is resistant to CsA, in the context of an intact core. Together, these results further support our model that CAeGFP is primarily located in vRNPs, and that the CsA-resistant CDR pool shown in (Fig 3E) is likely recruited by CAeGFP proteins inside an intact conical core.

## A fraction of unlabeled CA and CDR gets packaged with vRNPs

Our results (Figs 3 and S4) suggest that analysis of CDR retained by cores in the presence of CsA can faithfully report the packaging of tagged CA with vRNPs inside capsids. We next asked if unlabeled CA can also associate with vRNPs, and if this quantity can be estimated by incorporating CDR. To test this possibility, we incorporated INmNG vRNP-marker and CDR into chimeric HIV-1 capsids containing varied proportions of G89V and K203A CA mutants. Virions made by co-transfecting WT and CA mutant constructs were expected to recruit CDR in proportion to the intact CypA-binding loop (amino acids G89, P90) of the K203A CA mutant in the capsids. We reasoned that, following saponin and CsA treatment, the majority of virus-incorporated CDR will be released from capsids containing the unstable K203A CA, and only a subset of CDR will remain bound to unlabeled CA (if any) in the vRNPs (S5A Fig).

Imaging of single virus particles immobilized on poly-L-lysine treated glass revealed that, as expected, virions that contained G89V CA alone failed to efficiently incorporate CDR and showed marginal CDR colocalization (~2.5%) with the INmNG labeled cores. In contrast, mixed G89V/K203A CA virions efficiently incorporated CDR, showing 74–94% colocalization with INmNG labeled cores (S5B Fig). The dependency of CDR on the CypA-binding loop of K203A CA for its incorporation into mixed G89V/K203A capsids was further confirmed by fluorescence intensity analysis, which showed a correlated increase in virion-incorporated CDR fluorescence upon increasing the proportion of K203A CA plasmid in the transfection mixtures (S5C Fig). Addition of saponin released the majority of virus-incorporated CDR from the unstable mixed G89V/K203A capsids, and CsA-treatment released the remaining low-level of CDR from K203A vRNP complexes to near background levels (S5C Fig). Quantitation of the INmNG-associated CDR signals following saponin and CsA treatment relative to the total CDR incorporated into 100% K203A CA-containing virions revealed that as much as ~1% of virion-incorporated CDR (S5D Fig) can be incorporated into vRNPs. Based on the observation that one CDR molecule interacts with two adjacent CA molecules [49], we propose that as much as ~2% of virus incorporated CA is packaged in vRNPs.

## Unlabeled CA protein interacts with vRNP components

Our observation that eGFP-tagged CA and perhaps unlabeled CA can co-package with vRNPs (Figs 3 and S5) prompted us to test if vRNP-association might result from a direct interaction with CA. To address this possibility, we lysed native HIV-1 particles with 0.5% TX-100 and performed co-immunoprecipitation (co-IP) experiments with antibodies targeting vRNP components RT, IN, and NC. Additionally, antibodies against CA/p24 or Env glycoprotein were

used as positive and negative controls, respectively. In parallel, mock IP reactions were performed using the same antibodies in the absence of virus lysate to control for background detection of antibody protein chains. Immunoblots were probed with monoclonal anti-CA 183 or HIV-IgG antiserum to detect CA/p24 and viral proteins, respectively. As shown in Fig 4, IP of either RT, IN, or NC individually brought down the other two components of the vRNPs, and also a subset of CA/p24. Reciprocally, IP of CA pulled down a subset of input RT, IN, and NC proteins (Figs 4 and S6). By contrast, in control anti-gp120 antibody lanes, other viral proteins were not detected. Collectively, these results (observed in multiple experiments, n = 4) support the notion that CA interacts with vRNP components, suggesting a mechanism for its co-packaging with vRNPs during virus maturation, and for retaining eGFP-tagged and untagged CA with VRCs during nuclear import and subsequent nuclear trafficking.

## HIV-1 nuclear import is associated with capsid permeabilization

The inability of eGFP-tagged CA to accurately report uncoating prompted further investigation into the structural and/or compositional changes in capsids during HIV-1 nuclear import. We took advantage of the fact that eGFP-tagged CA or INmNG-CA packaged into vRNPs co-recruited CDR through low-affinity binding and released this CDR upon core uncoating *in vitro* (Figs 3E, S4 and S5D). This CDR pool was therefore used as a marker for loss of HIV-1 core integrity in cells. We reasoned that disruption and/or remodeling of the capsid lattice during nuclear import [17,18] should result in the loss of both CDR populations: CDR that is tightly bound to the CA lattice and CDR that is weakly bound to eGFP-tagged CA in vRNPs (S7A Fig). In contrast, CDR will remain entrapped and colocalized with eGFP-tagged CA or INmNG-CA during the nuclear import of intact capsids (S7A Fig). To test these alternatives, we infected TZM-bl cells with HIV-1 pseudoviruses co-labeled with CDR and eGFP-tagged CA or INmNG-CA, and examined the nuclear eGFP- or INmNG tagged VRCs for CDR signal at 4 hpi. Analysis of nuclear eGFP-tagged CA or INmNG-CA puncta revealed that CDR signal was markedly diminished in these complexes (S7B Fig). In contrast to nuclear VRCs, a significantly higher fraction of vRNP-associated CDR signal remained trapped within intact eGFP-tagged CA or INmNG-CA labeled cores on coverslips following saponin and CsA treatment *in vitro* (Figs 3D, 3E, S4D–S4F and S7C). Thus, HIV-1 nuclear import is associated with capsid permeabilization and release of the internal CDR marker.

As an alternative method to probe capsid permeabilization, we tested the accessibility of IN, which is part of the vRNP and resides inside capsids, during nuclear import. We used MT4 cells constitutively expressing the integrase binding domain (IBD) of lens epithelium-derived growth factor (LEDGF)/p75 fused to the C-terminus of the Nup153 nuclear envelope targeting cassette (NETC) [30]. As a control, MT4 cells expressing the NETC-GFP-IBD(D366N) mutant incapable of interactions with IN [30,50,51] were used. Confocal imaging revealed that only a fraction of nuclei expressing NETC-GFP-IBD were permissive for import of IN-labeled VRCs, whereas efficient nuclear import was observed in cells expressing the IBD(D366N) mutant (Fig 5B and 5C). NETC-GFP-IBD expressing cells contained >4-fold fewer VRCs per nucleus compared to NETC-GFP-IBD(D366N) expressing cells (Fig 5D), in excellent agreement with reduction in virus infectivity in the former cells (Fig 5E). The INsfCherry signal in NETC-GFP-IBD MT4 cells was observed at the nuclear envelope but not in the nucleoplasm, suggesting core integrity becomes compromised either before or during transit through nuclear pores. This result is consistent with the prior reports [30,31] and further demonstrates capsid permeabilization and the exposure of VRC-associated IN to the IBD at the nuclear pore.

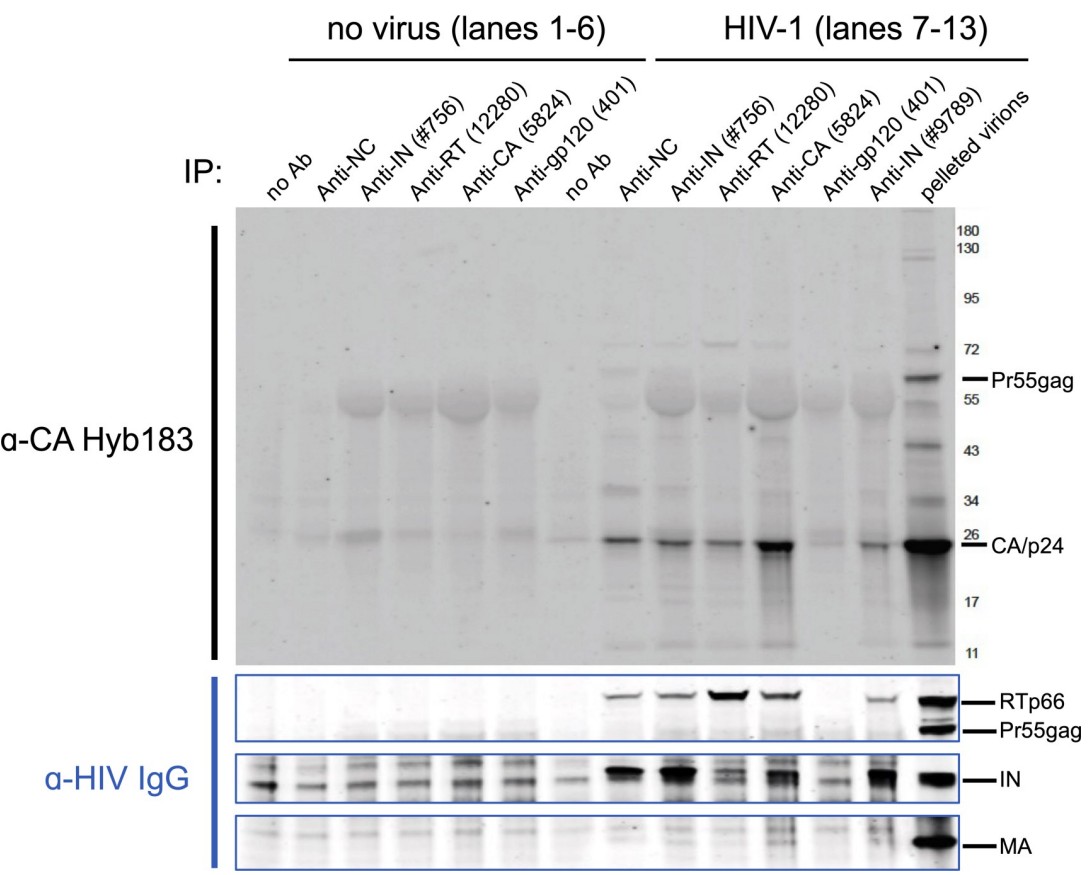

**Fig 4. A subset of native CA co-precipitates with HIV-1 vRNP proteins.** Virus lysates were subject to IP with antibodies against vRNP components IN, RT, and NC proteins. As controls, anti-CA and anti-gp120 envelope antibodies were used for IP. Mock IP was performed without virus lysate to control for possible secondary antibody bands. Immuno-blots were probed with anti-CA Hyb183 antibody (top) to detect CA/p24 or with HIV-IgG human serum (bottom) to identify HIV-1 proteins based on their respective molecular weight. Viral lysate (2.5 μg p24) was loaded in the last well to detect all viral components. HIV-IgG blots are cropped to show distinct RT, IN, and MA proteins. The fullsize blot is shown in S5 Fig. A representative immunoblot from 4 independent pulldown experiments is shown. In all experiments, a faint CA/p24 band co-immunoprecipitated with vRNP proteins.

## HIV-1 nuclear transport is highly sensitive to changes in the fraction of WT CA incorporated into mutant CA cores

Finally, we assessed the effect of the WT CA composition of HIV-1 cores on interactions with CPSF6, which is critical for intranuclear transport and integration site targeting [10,27,44,52]. Toward this end, variable fractions of unlabeled WT and mutant N74D CA, which does not bind CPSF6, were co-incorporated into INsfGFP-labeled viruses (Fig 6). In Fig 2, we co-mixed WT and N74D CA proteins in phenotypically mixed virions at the ratio of 1:10. To comprehensively address the role of CPSF6 binding in cytoplasmic restriction (*via* the CPSF6-358 truncation mutant) versus integration site targeting, we expanded this approach to span plasmid co-transfection ratios between 20:1 and 1:1. The minimal fraction of WT CA required for CPSF6 interactions was tested by probing the sensitivity of the resulting phenotypically mixed virions to restriction by the cytosolic CPSF6-358 fragment (Fig 2 and [41]) and by measuring the ability of endogenous CPSF6 to mediate nuclear transport and integration site targeting [10,27,44,52].

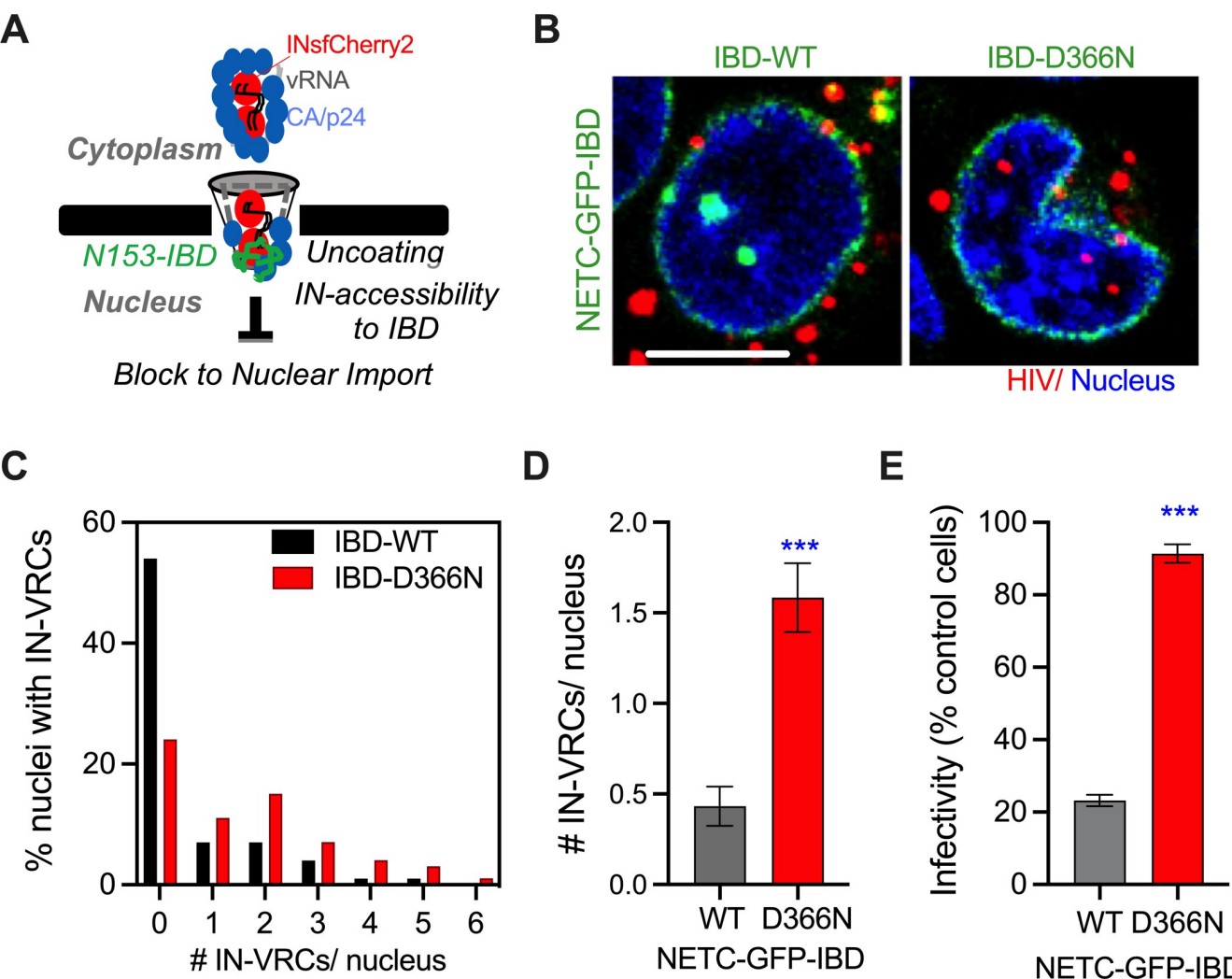

**Fig 5. HIV-1 IN becomes accessible at the nuclear pore prior to the nuclear import of VRCs.** Nuclear import and infectivity of VSV-G pseudotyped HIV-1 labeled with INmCherry (red) was tested in MT4 cells stably expressing the dominant-negative NETC-GFP-IBD wild-type (WT) and mutant D366N fusion proteins localized to the nuclear pore complex. **(A)** Illustration of nuclear pore-associated NETC-GFP-IBD protein binding to INsfCherry and blocking HIV-1 nuclear import. **(B)** Representative central Z-slice images of MT4 cell nuclei showing the localization of NETC-GFP-IBD constructs (green) and INsfCherry puncta (red). Nuclear INsfCherry VRCs are detected in cells expressing the mutant IBD-D366N (right), but less so in cells expressing the IBD-WT (left) fusion protein. Scale bar is 5 μm. **(C, D)** Quantification of the fraction of nuclei containing INsfCherry VRCs (C), and the average nuclear import (D) at 6 hpi. Shown are means and SEM from >120 nuclei from 4 independent experiments. **(E)** Infectivity measured at 72 hpi using a luciferase assay and normalized to control MT4 cells lacking the GFP-IBD proteins. Plotted are means and SEM from 4 independent experiments. Statistical analysis: non-parametric Mann-Whitney rank sum test (D) and student's t-test (E).

As expected, viruses with 100% WT CA were potently restricted in cells expressing CPSF6-358 (Fig 6A) and were detected in the nucleoplasm of control TZM-bl cells, >0.5 μm away from the nuclear envelope by 6 hpi (Fig 6B). By contrast, viruses containing only N74D CA were resistant to CPSF6-358 restriction in the cytoplasm and remained in the vicinity of the nuclear envelope in control cells (Fig 6A and 6B). Interestingly, the sensitivity to cytoplasmic CPSF6-358 restriction and VRC nuclear transport efficiency exhibited markedly different dependencies on the fraction of WT CA vs mutant CA plasmids used to transfect virus-producing cells. The CPSF6-358 fragment restricted infection by mixed WT/N74D CA viruses that contained as little as 5% of WT CA present in virus-producing cells (Fig 6A). In contrast to a dominant effect of small amounts of WT CA on recognition of cytosolic HIV-1 by CPSF6-

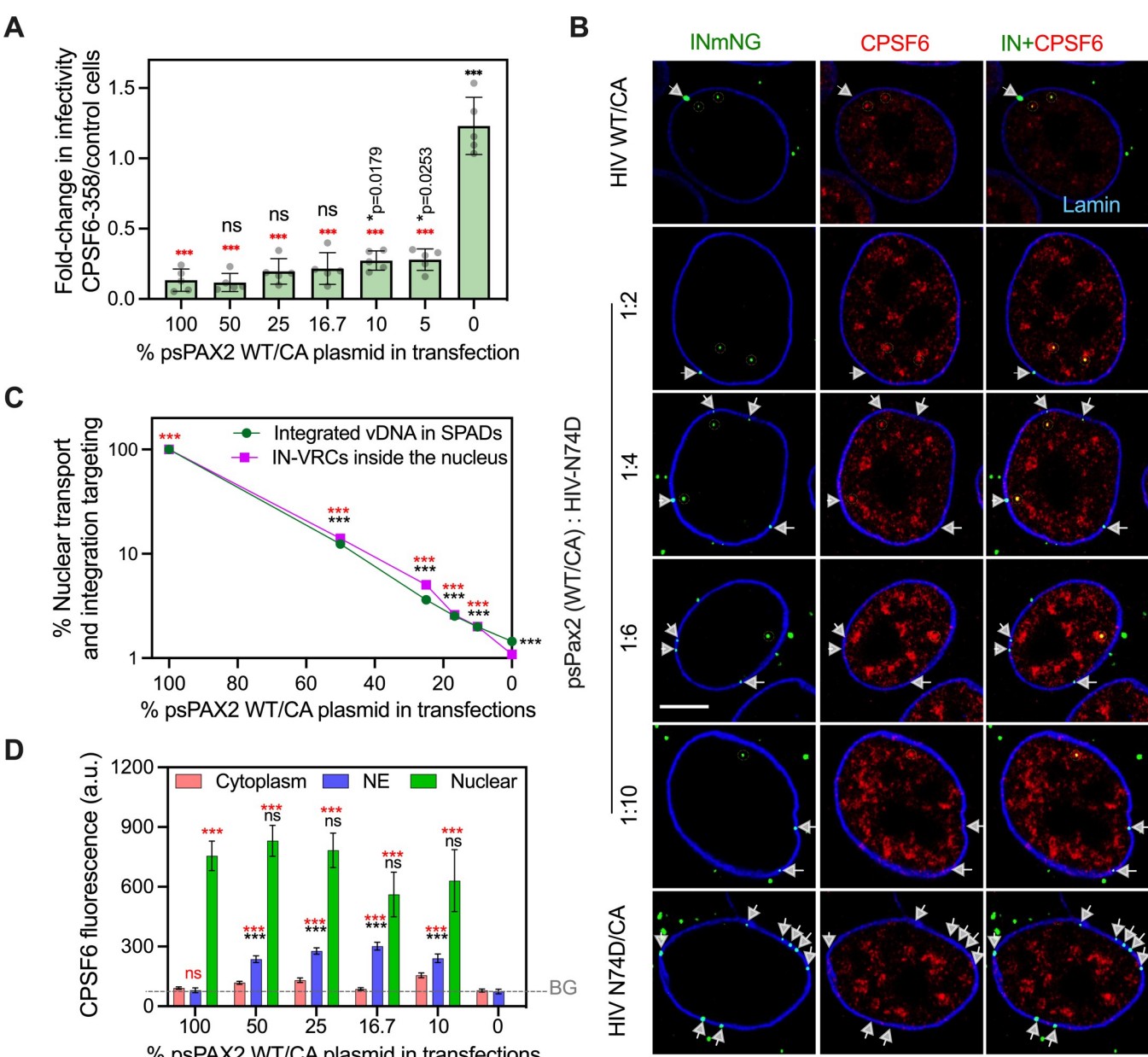

**Fig 6. HIV-1 nuclear import and integration targeting is associated with drastic changes to the structure and composition of the viral core. (A)** The ability of HIV-1 with mixed WT/N74D CA to interact with cytoplasmic CPSF6-358 was determined in TZM-bl cells. The ratio of infection in CPSF6-358 to control permissive TZM-bl cells is plotted. Here, 5% WT/CA incorporated into N74D/CA mutant cores was sufficient for cytoplasmic CPSF6-358 interactions. The average values from 5 independent experiments with SEM are shown. **(B)** Single Z-plane images showing IN-labeled HIV-1 VRCs (green puncta) colocalized with CPSF6 in the nucleoplasm (yellow dashed lines) and VRCs stuck at the nuclear envelope without CPSF6 recruitment (white arrowheads) in fixed TZM-bl cells at 6 hpi. Endogenous CPSF6 (red) and the nuclear lamin (blue) were detected by immunostaining. Scale bar is 5 μm. **(C)** The efficiency of nuclear penetration of IN-labeled VRCs (magenta squares/ line) for viruses with mixed CA or a 100% N74D/CA was determined by normalizing to the nuclear penetration of 100% WT/CA at 6 hpi in TZM-bl cells. Nuclear penetration was defined by fraction of IN-VRCs detected inside >0.5 μm of the NE with respect to the total nucleus associated HIV-1 complexes (NE + inside nucleus). The efficiency of SPAD-targeted integration (green circles/line) of the same viral preps was determined in 293T cells after 5 days of infection by normalizing to the SPAD-localized integration sites of a pure WT/CA virus. Nuclear penetration of VRCs and integration targeting into SPADs for viruses with a mixed WT/N74D CA decayed exponentially as a function of fraction of N74D CA. Data are means and standard deviations (too small to be visible) from 2 independent experiments for integration site analysis and 4 independent experiments for VRC penetration analysis. Data without normalization is shown in S7 Fig. **(D)** IN-associated CPSF6 signal recruited by VRCs in the cytoplasm, NE and >0.5 μm inside the nucleus. The background (BG) CPSF6 fluorescence determined in NE associated N74D-VRCs is shown as dashed grey line. Data is mean and SEM from 4 experiments, n>120 nuclei analyzed for each condition. Statistics in (A, C and D): non-parametric Mann-Whitney rank sum test in black *vs.* 100% WT/ CA control and in red *vs.* 100% N74D/CA.

358, WT CA fractions as high as 50% were insufficient to fully rescue nuclear transport of N74D-containing VRCs (Figs 6B, 6C and S8A). In agreement with these nuclear transport results, we observed an identically steep dependence of vDNA integration into SPADs on the WT fraction of CA within mixed WT/N74D viruses (Figs 6C and S8B). We do note that the few mixed WT/N74D HIV-1 VRCs that penetrated the nucleoplasm efficiently co-localized with SC35-labeled nuclear speckles (S9 Fig).

The inability of higher fractions of WT CA to fully rescue nuclear transport and integration site targeting of N74D viruses was not due to the lack of interaction with nuclear CPSF6 (Fig 6D). A subset of HIV-1 VRCs with mixed WT/N74D CA at the nuclear envelope and inside the nucleus showed a similar recruitment of endogenous CPSF6 to the virus with 100% WT CA (Fig 6D). Notably, in comparison with N74D viruses, both the efficiency of nuclear transport and SPAD integration targeting were significantly improved for VRCs with mixed WT/N74D CA, but remained markedly less efficient than 100% WT CA viruses (Fig 6C and S1 Table). Moreover, consistent with their proximity to the nuclear pore, the IN of mixed WT/N74D VRCs showed an increased susceptibility to the dominant negative IN-binding NETC-GFP-IBD fusion proteins located at the NPC (S10 Fig). Thus, unlike restriction by cytoplasmic CPSF6-358 (Fig 6A), the efficiency of nuclear transport and integration targeting was highly dependent on the composition of WT CA in HIV-1 VRCs, which were plausibly remodeled during nuclear import.

## Discussion

The process of HIV-1 uncoating remains enigmatic, in part due to a dearth of non-invasive direct CA labeling approaches to reliably visualize this process in the context of productive infection. HIV-1 CA is highly sensitive to genetic modifications [53], and insertions of small tags [54,55] can result in a drastic loss of infectivity. Even more recent labeling approaches that rely on the incorporation of single unnatural amino acids into capsids reduce infectivity and greatly delay virus nuclear import [56]. As a result of this "genetic fragility" [53], direct CA labeling approaches, such as the eGFP-tagged CA markers investigated here [7,29], rely on co-incorporation of an excess of unlabeled CA into virions to preserve infectivity. However, the precise nature of GFP-tagged CA association with the viral core remains unclear.

Here, we developed several approaches to localize eGFP-tagged CA in virions and to determine its post-infection fate and, thus, its utility as an image-based marker for HIV-1 uncoating. Consistent with the results of previous studies [7,29], our data (Figs 1 and 3) show that most of eGFP-tagged CA partitions outside of the assembled capsid and is released upon permeabilization of the viral membrane *in vitro*. Furthermore, our results reveal that the remaining core-associated eGFP-tagged CA proteins are: (1) largely excluded from the assembled conical capsid shell, as they fail to interact with the known CA-binding host-factors; and (2) packaged inside the HIV-1 core, apparently through interactions between CA and vRNP components. Our work therefore reveals that a subset of CA can be packaged with vRNPs, which should be considered while validating fluorescent CA markers for uncoating studies and highlights the critical need for developing new non-invasive capsid labeling approaches.

While our data (Fig 3) suggests that the majority of core-incorporated eGFP-tagged CA is vRNP associated in unstable K203A capsids, it does not rule out the possibility that a small sub-fraction of tagged-CA is incorporated into the assembled WT capsid. However, in that case, visualizing a complete loss of eGFP-signal in the nucleus is unlikely to reflect capsid uncoating, as these signals would be dominated by the majority of eGFP-tagged CA in the vRNP. As a result of vRNP association, the loss of eGFP-tagged CA inside the nucleus, which occurs concomitantly with loss of the IN marker (S3 Fig), is likely to occur during integration,

as evidenced by the inhibition of this loss by the IN inhibitor raltegravir. Our data disagrees with prior work that showed that nuclear eGFP-tagged CA disappeared in the presence of raltegravir [7]. The reason for this discrepancy remains unclear.

One of the arguments supporting the notion that nuclear eGFP-CA signal represents the CA lattice in intact HIV-1 cores is based on a loss of single eGFP-tagged CA spots in the nucleus upon treatment with 10 μM PF74 [7]. The rationale is that PF74 binds to the intra-hexamer pocket of a CA lattice and destabilizes intact cores. It is noteworthy, however, that loss of HIV-1 fluorescent puncta in the nucleus induced by a higher concentration of PF74 (25 μM) occurs as a result of their exit from nuclear speckles, and not due to a direct destabilizing effect of PF74 on the viral core [10]. On the contrary, we have observed that PF74 (10–25 μM) stabilizes HIV-1 cores in the cytoplasm of living cells, as well as *in vitro* [10,27,34]. This stabilizing effect of 10 μM PF74 on HIV-1 cores was also confirmed by other laboratories [35,57–59]. In agreement with this notion, 10 μM PF74 has been demonstrated to stabilize the CA lattice *in vitro*, while accelerating the initial steps of core opening [60]. We therefore caution against interpreting PF74-induced disappearance of eGFP-tagged CA puncta in the nucleus as evidence for intact capsid cores reaching these compartments.

We have previously used CDR as a high-avidity CA binding reporter to study uncoating and consistently observed that loss of CDR at the NPC was a prerequisite for HIV-1 nuclear import and infection across multiple cell types [6,10,27]. Here, we find that avid interactions between CDR and capsid are exclusive to the CA lattice, as this marker only weakly interacts with vRNP-associated INmNG-CA or eGFP-tagged CA (Figs 3 and S4). Based on these new results, we propose that, in addition to compositional changes during HIV-1 nuclear import, remodeling of the CA lattice will likely involve local rearrangements (structural changes), resulting in a loss of avid CDR-CA interactions at the NPC [27]. Loss of CDR from eGFP-CA and INmNG-CA tagged VRCs in the nucleus may be explained by dynamic changes in IN oligomerization observed during HIV-1 nuclear import [46].

Our results agree with the notion that some form of uncoating/remodeling precedes HIV-1 nuclear import and are inconsistent with the interpretation that cores remain intact and lost integrity only once inside the nucleus, as concluded recently [25]. In this work, Li and co-authors [25] made use of a Gag-iGFP marker that is inserted between the MA and CA domains of Gag, which is released upon viral protease cleavage during maturation. The iGFP is a readily releasable, fluid-phase marker inside virions, and a subset of this marker appears to be trapped in intact cores [26,61]. The release of iGFP marker occurs in two-steps: first, during virus-cell fusion and, then, upon the loss of capsid integrity [26]. While prior work [26] has observed a correlation between the cytoplasmic loss of iGFP and HIV-1 infection, Li *et al.* [25] reported that cores retain the iGFP marker in the nucleus and lose this marker during nuclear uncoating. These discrepancies likely reflect a heterogenous nature of iGFP incorporation into cores. It is currently unclear if all the residual iGFP marker trapped inside a conical capsid is free (fluid-phase) to be released upon capsid permeabilization during nuclear import. Future work will address this concern.

Given our observation that the loss of GFP-tagged CA signals in the nucleus correlates with integration rather than uncoating, the sites and extent of CA loss during HIV-1 nuclear import remain unclear. Here, in addition to the previously implemented CDR-based uncoating assay [6,10,27,34], we employed alternative approaches to probe the permeability of cores during nuclear import. Our results (Figs 5 and S7) show that: (1) IN becomes accessible to the NET-C-IBD construct targeted to the nuclear pore of MT4 cells, in agreement with previous reports [30,31], and (2) loosely packed, vRNP-localized CDR is lost upon HIV-1 nuclear import. At the same time, our data (Fig 6) confirm and expand the understanding of the role of CA in VRC nuclear transport to integration sites near nuclear speckles. Strong dependence of these

nuclear processes on multiple CA-CPSF6 interactions in mixed WT/N74D CA complexes highlights the critical role of CA in the nucleus. The marked difference between the fraction of WT CA required to rescue restriction by CPSF6-358 in the cytoplasm vs the fraction that effectively directs integration to nuclear SPADs might reflect progressive capsid disassembly and/ or remodeling of capsids, consistent with prior reports [17,18].

As an initial test of vRNP-associated CA functionality, we incorporated wild-type CA at the C-terminus of Vpr-INmNG (Vpr-INmNG-CA, S4 Fig) and assessed the ability of this novel fusion protein to direct N74D virus integration into SPADs. As shown in (S2 Table), the basal levels of SPAD targeted integration for the unlabeled WT and N74D viruses was 29.7% and 0.7%, respectively, and these levels remained largely unaffected by the presence of vRNP-associated INmNG-CA proteins. These results suggest the inability of INmNG-CA proteins to associate with CPSF6 and mediate integration targeting. There could be several reasons why CA fused to INmNG failed to appreciably stimulate SPAD-tropic HIV-1 integration targeting (S2 Table). While CA hexamerizes to form the intra-hexamer CPSF6 binding pocket, functionally relevant IN assemblies are four-fold in nature (tetramer and hexadecamer [13,14,62]). Thus, the propensity for IN to tetramerize could negate the ability of fused CA moieties to functionally hexamerize. Otherwise, CA may need to be part of the mature capsid lattice to effectively engage CPSF6 for integration site targeting. We surmise that this assembled CA which interacts with CPSF6 in the nucleus originates from a subset of the CA lattice that remains following progressive disassembly/ remodeling of the capsid during virus ingress and nuclear transport [1,35].

How partial CA lattices are retained by VRCs in the nucleus remains unclear. In this context, an important finding of the current study is the ability of CA to interact with vRNP components (Fig 4). Given that HIV-1 DNA synthesis is completed in the nucleus [6–8,10,63], at least some vRNP components, perhaps RT and IN, may be involved in retaining CA in nuclear VRCs. Additionally, it will be interesting to probe if CA/vRNP-interactions are facilitated by host-proteins packaged into virions, and if such interactions are retained in the nucleus. Our findings that CA interacts with vRNP components may have broad implications in HIV-1 biology. We surmise that the interactions between CA and vRNP, which are likely initiated during virion morphogenesis, may play a role in proper assembly, maturation and vRNP-encapsidation inside a conical capsid shell. Intriguingly, vRNP encapsidation is universally disrupted by the large subset of IN mutations that are collectively referred to as class II IN mutation [64,65]. Future research will aim to map CA domains/residues that mediate vRNP binding, and the vRNPs components responsible for CA recruitment and retention in the nucleus.

## Materials and methods

### Plasmids

The pR9ΔEnv with CA wild-type or mutants G89V and K203A [47], NL4.3 R-E- Luc [66] have been described. Env-defective HIV-1 GFP reporter proviral constructs encoding the CA substitutions G89V and N74D CA were constructed by transfer of BssHII-ApaI fragments encoding the corresponding mutations from R9 plasmids into pHIV-GFP [67]. Constructs were verified by Sanger sequencing of the transferred regions. The pMD2.G vector (Addgene plasmid # 12259), and the psPAX2 vectors (Addgene plasmid# 12260) were obtained from Addgene and was a kind gift from Didier Trono (EPFL, Switzerland). The HGFP-CA vector encoding N-terminal GFP-tagged CA was a kind gift from Dr. Vinay Pathak (NCI, [7]). The GagIN-GFP (GIG) plasmid [35] was a kind gift from Dr. Tom Hope (North Western University, Chicago IL). The pcDNA3.1-MA-CAeGFP construct was made as follows: the MA-CA

sequence was PCR amplified from GagIN-GFP and cloned into a pcDNA3.1 vector between KpnI-BamHI sites. Subsequently, a PCR amplified in-frame eGFP sequence was introduced into the MA-CA construct at BamHI and NotI sites to make the C-terminally tagged pcDNA3.1 MA-CAeGFP construct. The CypA-DsRed-Max (CDR), Vpr-IN-superfolderGFP (INsfGFP), and Vpr-INmNeonGreen (INmNG) expressing plasmids have been described previously [27,34,68]. The pSFFV_sfCherry2_TagBFP [69] was a gift from Bo Huang (Addgene plasmid # 83031). The sfCherry2 from this plasmid was PCR amplified and cloned into the Vpr-INmNG plasmid by replacing mNG using BamHI and NotI sites by molecular cloning to make the Vpr-INsfCherry2. The Vpr-INmNG-CA chimeric construct was made as follows: the INmNG sequence without a stop codon was PCR amplified with primers containing HindIII/ AgeI overhangs at the 5' and 3' ends; similarly, the CA sequence starting from PIVQN and ending in a stop codon (TAA) was amplified from the MA-CA construct using primers containing AgeI and NotI overhangs at the 5' and 3' ends. Subsequently, after enzyme digestion with HindIII, AgeI or AgeI and NotI sites of respective PCR products, they were cloned by a 3-way ligation into Vpr-INmNG construct by replacing INmNG. All plasmid sequences were verified by Sanger sequencing (Genewiz, USA).

## Cell lines and reagents

HEK293T/17 cells (from ATCC, Manassas, VA), HeLa-derived TZM-bl cells (from NIH AIDS Reference and Reagent Program) and TZM-bl cells expressing SNAP-LaminB1-10 (Francis et al 2020) or TRIMCyp WT or the TRIMCyp126 mutant [39] (kind gift from Dr. Jeremy Luban, University of Massachusetts Medical School) or the cytosolic CPSF6-358 [41] (kind gift from Dr. Vineet Kewalramani, NCI) were grown in complete high-glucose Dulbecco's Modified Eagle Medium (DMEM, Mediatech, Manassas VA) supplemented with 10% Fetal Bovine Serum (FBS, Sigma, St. Louis, MO) and 100 U/ml penicillin-streptomycin (Gemini Bio-Products, Sacramento, CA). The growth medium for HEK293T/17 was supplemented with 0.5 mg/ml G418 sulfate (Mediatech, Manassas VA). MT4 cells stably expressing the NETC-eGFP-IBD WT and the D366N mutant [30] were maintained in RPMI-1640 complete medium (Mediatech, Manassas VA) supplemented with 10% Fetal Bovine Serum and 100 U/ml penicillin-streptomycin.

Cyclosporin A (CsA) was obtained from Calbiochem (Burlington, MA), dissolved in DMSO at 50 mM, and stored in aliquots at -20˚C. The Bright-Glo luciferase assay kit was from Promega (Madison, WI). Puromycin was obtained from InvivoGen. Primary antibodies to rabbit anti-mCherry antibody (1:500 dilution, #ab167453), Lamin-B1 (#ab16048) and a secondary donkey anti-rabbit AF405 antibody (#ab175651) were purchased from Abcam (San Francisco, CA). A primary antibody to CPSF6 (Rabbit PA5-41830, ThermoFisher), the Cy5-conjugated anti-mouse secondary antibody was from SouthernBiotech (Birmingham, AL). The SNAP-Cell 647-SiR dyes was purchased from New England Biolabs (NEB, #S9102S). Phosphate buffered saline containing $Mg^{2+}/Ca^{2+}$ (dPBS) and Mg/Ca-free (PBS) were purchased from Corning (MediaTech, Manassas, VA). Hoecsht33442 was from ThermoScientific (#62249). Precast gel 4–20% (Cat# 4561091) and pre-stained precision plus protein markers (#1610374) were from BioRad (CA, USA).

The following reagents were obtained from the NIH HIV Reagent Program, Division of AIDS, NIAID, NIH: pNL4-3.Luc.R-E- from Dr. Nathaniel Landau [66,70], TZM-bl cells expressing CD4, CXCR4 and CCR5 from Drs. J.C. Kappes and X. Wu [71]; anti-p24 antibody AG3.0 donated by Dr. J. Alan [72]; RT inhibitor Nevirapine and IN inhibitor Raltegravir (Merck & Company). The following antibodies were obtained from the NIH AIDS Reagent Program, Division of AIDS, NIAID: monoclonal antibody to HIV-1 CA (produced from

hybridoma 183-H12-5C, ARP#1513), rabbit anti-HIV-1 RT (ARP#12280), rabbit anti-HIV-1 IN (ARP#9789), rabbit anti-HIV-1 IN (ARP#756), HIV-1$_{HXB2}$ Integrase antiserum (aa 276–288) from Dr. Duane Grandgenett (cat. #758); rabbit anti-HIV-1 MA (ARP#4811); and HIV-IgG pooled human serum (ARP#3957). Rabbit polyclonal anti-gp120 was purchased from Intracel (cat. #401), goat polyclonal anti-HIV-1 NC was contributed by Dr. Robert Gorelick, National Cancer Institute, NIH; rabbit anti-CA, produced against full-length HIV-1 CA protein, was obtained from Didier Trono.

## Pseudovirus production and characterization

Fluorescently labeled pseudoviruses were produced and characterized, as described previously [34] using the JetPrime Transfection reagent (VWR, Radnor, PA) according to the manufacturer's protocol. To make the eGFP-tagged CA labeled viruses, the HIVeGFP backbone (2 μg), VSV-G (0.5 μg), MA-CAeGFP or the HGFP-CA (0.5 μg) plasmid were transfected at 10:1:1 ratio. Where indicated, mixed CA viruses were produced by co-transfecting 0.5 μg of VSV-G plasmid with 0.2, 0.33, 0.5 or 1 μg of WT Gag/GagPol encoding psPax2 vector and with 1.8, 1.67, 1.5 or 1 μg of Env-deleted HIVeGFP proviral plasmid encoding for the mutant N74D or G89V CA. These plasmid ratios were used to produce viruses nominally containing 10, 16.7, 25 and 50% of WT CA relative to the mutant CA. When co-labeling with a second fluorescent protein, 0.8 μg of the Vpr-INsfCherry2 or 0.5 μg of CypA-DsRed (CDR) plasmids was included in respective transfection mixtures. Viruses co-labeled with IN-mNG-CA and CDR were produced by co-transfecting 2 μg of HIVeGFP WT or G89V CA backbone with 0.5 μg of VSV-G, 0.8 μg of Vpr-INmNG-CA and 0.5 μg of CDR plasmids.

After 6 hours of transfection, the medium was replaced with 2 ml of fresh complete DMEM without phenol red. After further incubation for 36 h at 37˚C, 5% CO$_2$, viral supernatants were collected, filtered through a 0.45 μm filter and quantified for the RT activity (RTU) measured using the SG-PERT protocol, as described previously [27,73]. For live-cell imaging, fluorescent viruses were concentrated 10x using Lenti-X concentrator (Clontech Laboratories, Inc. Mountainview, CA). Concentrated viruses were re-suspended in complete FluoroBrite (GIBCO) or RPMI-1640 medium or PBS, aliquoted and stored at -80˚C. The MOI was determined in TZM-bl cells by examining the % of eGFP expressing cells after 48 hours post-infection with VSV-G pseudotyped HIVeGFP virus.

## Single-round infectivity and restriction assay

Virus infectivity was measured by a luciferase reporter expression in TZM-bl cells or TZM-bl cells expressing the restriction-competent host factors TRIMCyp or CPSF6-358. Ten thousand TZM-bl cells were plated in triplicate wells of a 96-well plate and infected with VSV-G pseudotyped HIV-1 viruses (MOI 0.2–0.5), as indicated in text. Virus binding to cells was enhanced by a 30 min centrifugation at 1,500×g, 16˚C. Cells were cultured at 37˚C for additional 48 h, lysed, and the luciferase activity measured using the Bright-Glow luciferase substrate (Promega). For image-based quantification of infected cells, 4x10$^4$ TZM-bl cells in a 8-well chambered coverslip were infected with unlabeled or HIVeGFP-labeled viruses/ The % infected cells was determined by counting the number of eGFP reporter expressing cells from 4 random fields of view and normalizing to the total number of cell nuclei determined by Hoechst staining. When performing restriction assays to determine CA/host-factor interactions, cells expressing the restriction competent TRIMCyp322 or CPSF6-358 proteins and control cells expressing TRIMCyp126 or endogenous full-length CPSF6 proteins, respectively, were infected. Serial 10-fold dilutions ($10^{-1}$–$10^{-4}$) of normalized un concentrated viral supernatant containing 0.2–0.5 RTU were used in infection. The luciferase signal (RLUs) measured at 48

hpi in restriction competent cells was normalized by dividing over the RLUs from control cells, and the ratio was used to determine the efficiency of host factor binding and restriction.

## Immunoprecipitation and western blotting

HIV-1 particles unlabeled or labeled with eGFP-tagged CA proteins were concentrated 10x times in dPBS (Ca+/ Mg+; Corning MediaTech, Manassas, VA) using the Lenti-X reagent. Concentrated viral suspension equivalent to 2.5 RTU was loaded on a 4–20% pre-cast poly-acrylamide gel. The separated proteins were transferred onto a nitro-cellulose membrane, blocked with 2.5% skimmed milk in PBST (PBS+0.1% tween 20), and incubated with HIV-IgG human serum (1:2000) or Living colors monoclonal antibody to GFP (1:200) for 1h at room temperature or overnight at 4˚C. After three 10-minute washes with PBST, the membrane was incubated with secondary anti-human HRP (Abcam, cat# ab6858) (1:10000) or anti-mouse HRP (Azure biosystems, cat# AC2115) (1:1000) antibodies. The membrane was developed in ECL solution (Amersham, cat# RPN2209) for 2 min prior to imaging on a BioRad gel imager.

Immunoprecipitations shown in Fig 4 were performed by diluting concentrated HIV-1 particles (2.1 μg of p24, produced by transfection of 293T cells with the R9 molecular clone) into 0.5 ml of cold STE buffer (10 mM Tris-HCl pH 7.4, 100 mM NaCl, 1 mM EDTA) containing 0.5% Triton X-100 and the indicated antibodies. Reactions were incubated for 1.5 h at 4˚C, then 30 μl of prewashed Protein A/G+ bead suspension (Santa Cruz, Inc) were added and the tubes rotated end-over-end for 1h. Beads were washed twice in STE+0.5% Triton X-100. Proteins were eluted and denatured by resuspending the pelleted beads in 30 μl of Laemmli buffer and heating at 95˚C for 5 min. Eluted proteins were subjected to electrophoresis on 4–20% SDS-polyacrylamide gels (Genscript). A reference sample corresponding 600 ng of lysed virions was also loaded on the gel. Proteins were transferred electrophoretically to nitrocellulose membrane, which was subsequently blocked and probed with anti-CA (1 μg/ml) followed by HIV-Ig (1:10,000 dilution in PBS containing 3% BSA). Bands were visualized by probing with IR dye-conjugated commercial secondary antibodies and scanning with a Li-COR Odyssey imaging system.

## Single virus imaging *in vitro*

Fluorescently labeled HIV-1 viruses were bound to a poly-L-lysine treated #1.5 8-well chambered coverglass (Thermo Fisher Scientific, cat#155409,) for 30 min at 4˚C. After a 1x wash with dPBS (Ca+/Mg+), bound viruses were imaged on a Zeiss LSM880 or Leica SP8 confocal microscopes using a C-Apo 63x/1.4 N.A. oil objective. Imaging was performed in dPBS using the same parameters as for fixed cells described below. Four fields of view were imaged using the 488 and 561 nm laser lines, and respective emissions for eGFP (502–550 nm) and DsRed or sfCherry2 (572–640 nm) were collected with PMTs or HyD detectors. For analysis of HIV-1 core-associated fluorescence, the membrane of glass-bound viruses was permeabilized with saponin (100 μg/ml) for 30s, and imaging was continued to monitor the loss of eGFP or DsRed fluorescence. Where noted, CsA (10 μM) was added to permeabilized virions 1 min after saponin to displace CDR from HIV-1 cores.

## Immunofluorescence and fixed cell imaging

For fixed cell imaging, $1\times10^6$ MT4 cells expressing NETC-GFP-IBD or $5\times10^4$ TZM-bl were infected in a 8-well chambered coverslip at MOI 0.2 with fluorescently tagged viruses using spinoculation, as described above. Cells were washed and incubated at 37˚C in 5% $CO_2$ for indicated time intervals. Cells were fixed with 2% PFA (Electron Microscopy Sciences, #1570-S) for 7 min at room temperature. MT4 cells were pelleted and adhered to a 8-well

chambered poly-D-lysine coated coverglass for 30 minutes at 4˚C. Fixed cells were either directly imaged in dPBS or further processed for immunofluorescence as follows: PFA-fixed cells were permeabilized with 0.1% TX-100 for 5 min at room temperature, washed 3 times in PBST, and blocked with 3% BSA with 0.1% Tween-20 in dPBS. Primary anti-LaminB1 antibody (1:1000), anti-CPSF6 antibody (1:200) diluted in a blocking solution were allowed to bind for 1 h at room temperature or overnight at 4˚C. Cells were washed 5 times with PBST and incubated with secondary goat anti-mouse Cy5-conjugated antibodies (1:1000), washed 5x, and incubated with goat anti-rabbit-AlexaFluor405 (1:1000), each for 1 h at room temperature. Following a 5x wash in PBST the cells were imaged in dPBS on a confocal microscope.

## Live-cell imaging of HIV-1 infection

Single HIV-1 infection in live cells was visualized, as previously described (Francis and Melikyan 2018; Francis et al 2020b). In brief, $5 \cdot 10^4$ TZM-bl cells in a 8-well chambered slide were infected (MOI 0.2) with a VSV-G pseudotyped HIVeGFP particles co-labeled with INsfCherry2 and eGFP-tagged CA by spinoculation (1500×g for 30 min, 16˚C). Where mentioned, TZM-bls expressing the SNAP-Lamin nuclear envelop marker were labeled for 30 min with SNAP-Cell 647-SiR dyes (NEB, #S9102S) prior to virus binding. Following spinoculation, the cells were washed twice, and virus entry was synchronously initiated by adding prewarmed complete FluoroBrite medium to samples mounted on a temperature- and $CO_2$-controlled microscope stage. Where noted, the integrase inhibitor Raltegravir was included in live-cell imaging experiments at a final concentration of 10 μM. 3D time-lapse live cell imaging was carried out on a Zeiss LSM880 laser scanning confocal microscope, using a C-Apo 63x/1.4NA oil-immersion objective. Tile-scanning was employed to visualize uncoating, nuclear entry and infection of HIV-1 from 25 neighboring fields of view. Live-cell imaging was performed starting from 0.5 or 2 hpi to 22 hpi by acquiring 11–15 Z-stacks spaced by 0.7 μm every 5 min. The DefiniteFocus module (Carl Zeiss) was utilized to correct for axial drift. The SiR-SNAP-lamin stained nuclear membrane and CAeGFP or HGFP-CA, INmNG, INmNG-CA and INsfCherr2 or CypA-DsRed were imaged using highly attenuated 488, 561 and 633 nm laser lines. 3D-image series were processed off-line using ICY image analysis software (http://icy.bioimageanalysis.org/) [74].

## Image analyses

For assessing the CAeGFP, HGFP-CA, CDR, INsfCherry2, and immuno-labeled CPSF6 signals in the cytoplasm, NE and nucleus, stringent imaging conditions were used compared to those used for long-term time-lapse imaging (2x laser power, 4-line averaging, with 0.12 μm/pixel in X-Y and 0.3 or 0.5 μm/pixel in Z). 3D imaging of fixed cells was carried out on a Zeiss LSM880 or a Leica SP8 laser scanning confocal microscopes, using a C-Apo 63x/1.4NA oil-immersion objective. To discriminate between the cytoplasmic, NE and nuclear IN-spots, an in-house script was created using the ICY protocols module, essentially as described in [10]. Briefly, the nuclear volume in three dimensions was detected using the lamin intensity by the HK-means and the Connected Components plugins in ICY [75]. The obtained 3D ROI corresponding to the nuclear volume was shrunk by 0.5 μm in X-Y-Z using an ROI-erosion plugin. HIV-1 complexes detected within the eroded ROI were considered as nuclear spots. The nuclear lamin ROI was dilated by 0.5 μm in X-Y-Z using an ROI-dilation plugin, HIV-1 complexes detected in this dilated NE-ROI was considered NE associated. The total HIV-1 ROIs from the 3D cellular volume was then determined, and, after subtraction of NE- and nucleus-associated ROIs, the remaining HIV-1 spots were deemed to reside in the cytoplasm.

For intensity analysis, the ROI of the detected INsfCherry2-spots was expanded (dilated) by 1 pixel to pick up the local background (BG) for each punctum. The average intensity in the dilated-1-pixel region was calculated and subtracted from the average intensity of real ROIs. The BG correction routine was used for all intensity-based analysis. The BG-corrected intensity of CAeGFP, HGFP-CA, CDR associated with the INsfCherry2 or INmNG-CA reference signals was measured in cores *in vitro* or in nuclear VRCs. The fraction of IN spots that contained above-background levels of CAeGFP, HGFP-CA, CDR were considered co-localized with these markers. The absolute background was calculated by measuring the fluorescence signal associated with single HIV-1 labeled with IN alone from parallel infections.

Live-cells: The initial annotation of HIV-1 infected cells expressing the eGFP-reporter at the end of imaging was done manually by examining the time-lapse movies. After visual inspection, software-assisted single particle tracking was performed using ICY to analyze viral complexes entering the nucleus and determine the time of subsequent particle disappearance. Single particle intensity traces were normalized to initial fluorescence intensity of INmNG and CypA-DsRed after background subtraction. The nuclear SNAP-Lamin signals were normalized by subtracting the background signal and setting the peak intensity at the mid-lamin section to 100%. The nuclear entry and single complex disappearance times were determined by visual inspection for those events that were not amenable to single-particle tracking.

## Analysis of HIV-1 integration into SPADs

HIV-1 integration libraries were prepared by performing ligation-mediated PCR (LM-PCR) as described previously [76]. In short, genomic DNA (2–10 µg) from infected cells was digested with restriction enzymes MseI and Bgl II overnight at 37˚C. Digested and purified genomic DNA fragments were ligated with double stranded asymmetric DNA linkers overnight at 12˚C. These linkers contain restriction enzyme compatible TA overhangs at the 5'. The viral-host integration junction was amplified by nested PCR with purified ligated products as templates as described previously. Linker primers and the second round LTR primers contain Illumina specific sequences at their 5' end which is required for proper clustering during sequencing. Purified PCR products were multiplexed, clustered in 1 lane of the flow cells and subjected to 150 bp paired end sequencing on HiSeq-4000 at Genwiz.

Illumina raw reads were processed to determine individual integration sites as per previously described methodologies[77,78]. In short, HIV-1 U5 and linker sequences were identified and trimmed from read1 and read2, respectively. Trimmed read were aligned to human genome hg19 by BWA-MEM with paired-end option[79]. Aligned reads were filtered to remove unmapped, low quality reads and reads mapped to multiple location in the genome by SAMtools[80]. Filtered reads were processed by custom Python script to determine integration sites in BED format as described previously [78,81]. Integration sites were analyzed by BEDtools to assess their distribution against various genomic features [82].

Speckles-associated genomic domains (SPADs) were determined and defined previously [10,78] from TSA-seq data and method described by Chen et al.[83]. P values were calculated by Fisher's exact test in a pairwise manner using a Python script except in case of gene-density where Wilcoxon rank sum test was used to calculate P-values. Random integration controls (RICs) were calculated *in silico* previously [78], following digestion of hg19 with MseI and BglII restriction enzymes to control for biases in integration preference introduced by restriction enzymes.

## Statistical analysis

Statistical significance was determined by the non-parametric Mann-Whitney rank-sum test or the Student t-Test, as indicated. $p < 0.05$ (*) was considered significant; ** and *** denote

p<0.01 and p<0.001, respectively. The number of experiments and error bars are indicated in the figure legends.

## Supporting information

**S1 Fig. (A)** Schematics of C-terminally tagged CAeGFP in the context of full-length Gag/Gag-Pol reported in *(Zurnic et al, J. Virol 2020)*. **(B)** Fluorescence intensity analysis of eGFP-tagged CA signals in nuclear VRCs at 4 and 12 hpi was determined in TZM-bl cells, as reported in Fig 1G and 1H.
(TIFF)

**S2 Fig. GFP-tagged WT/CA is not exposed on the HIV-1 core for host-factor interactions.**
**(A)** SDS-PAGE of virus lysates (1ng of p24) of indicated HIV-1 CA WT and mutant viruses mixed with 10% of CAeGFP or unlabeled psPax2 CA probed with an anti-HIV-1 IgG (top) and anti-eGFP antibodies (bottom). Immunoblot shows efficient processing of MA-CAeGFP in virions (top) resulting in appearance of CAeGFP band (bottom). Putative bands corresponding to MA-CAeGFP, Gag/pr55 precursor, CAeGFP, and CA/p24 are marked. Note, in top HIV-IgG blot the processed CAeGFP (51 kDa) co-migrates with the prominent Pr55Gag band. **(B)** Strategy to test the localization of GFP-tagged CA in virions. HIV-1 containing G89V mutant CA are labeled with GFP-tagged WT/CA protein, and the ability of GFP-tagged CA to bind cytosolic TRIMCyp restriction factor constitutively expressed in target TZM-bl cells. The following scenarios are illustrated: *(i)* The G89V/CA mutant virus does not interact with TRIMCyp and naturally escapes restriction. *(ii)* If the virus-incorporated GFP-tagged WT/CA co-assembles on the capsid lattice, these proteins will interact with TRIMCyp and restrict G89V infectivity. *(iii)* If GFP-tagged CA is inaccessible to TRIMCyp, likely due to packaging into vRNP, G89V viruses will not be restricted. **(C)** Single-round infectivity data shows that GFP-tagged CA is inaccessible to TRIMCyp, while 10% unlabeled WT/CA in G89V viruses is recognized and restricted by TRIMCyp. The control HIV-1 infections with 100% unlabeled WT/CA or MT/CA were completely susceptible or escaped TRIMCyp restriction, respectively. Infectivity was normalized to control cells expressing the TRIMCyp126 mutant which does not interact with CA. The means and STDs from 4 experiments is shown. Statistics: in (C) student's t-test was used.
(TIFF)

**S3 Fig. GFP-tagged CA proteins co-disappear with INsfCherry labeled nuclear VRCs prior to expression of GFP reporter of infection.** TZM-bl cells constitutively expressing SNAP-lamin nuclear membrane marker were infected (MOI 0.2) with HIV-pseudoviruses co-labeled with indicated GFP-tagged CA proteins and INsfCherry2 marker. Live-cell imaging was performed starting from 1 h and continued up to 22 hpi. Infected cells were identified by the expression of eGFP reporter and analyzed for nuclear INsfCherry2/GFP-tagged CA complexes, and these single viral complexes were tracked. **(A, C)** Images and **(B, D)** single virus fluorescence trace of examples shown in respective images. The images in (A, C) show HIV-1 complexes entering the nucleus (dashed yellow circles) trafficking for several hours and co-disappearing (arrowhead) followed by eGFP reporter expression. Fluorescence traces in (B, D) show that eGFP-tagged CA signal remains steadily associated with INsfCherry2 puncta and co-disappears at 13 h 21 min (A, B) and 12 h 24 min (C, D). Scale bar in A, C is 2 μm. Quantification of: **(E)** HIV-1 puncta that disappeared in the nucleus of infected cells and contained both CA- and IN- signals, and **(F)** the fraction of cells showing eGFP/sfCherry puncta disappearing in the nucleus by 22 h in control (CNTRL) non-treated or in the presence of integration inhibitor raltegravir (+RAL, 10 μM). Data is cumulative from 2 independent 22 h live-

imaging experiments and (n) number of cells analyzed is shown.
(TIFF)

**S4 Fig. Selective localization of CA-proteins and CDR with vRNPs through INmNG-CA fusion proteins. (A)** Schematics of the chimeric INmNG-CA fusion construct, the protease cleavage site (PC: IRKVL/FLDGI) between Vpr- and IN is included to release INmNG-CA from Vpr after virion incorporation. **(B)** Immuno-blot analysis of unlabeled and INmNG or INmNG-CA labeled HIV-1 using anti-HIV-IgG antibodies. The migration of Vpr-INmNG-CA and processed INmNG-CA is marked. **(C)** Single round-infectivity of INmNG or INmNG-CA labelled HIV-1 with respect to unlabeled HIV-1 was measured using a luciferase assay in TZM-bl cells at 48 hpi (MOI 0.5). Dots represent independent triplicate experiments. **(D)** Schematic for the incorporation of CDR selectively with vRNPs in G89V CA mutant virus through co-incorporation with INmNG-WT CA fusion proteins. **(E)** Images showing lack- or efficient-labeling of G89V virus with CDR in the presence of INmNG control or INmNG-CA fusion protein, respectively. **(F)** Quantification of CDR signal in INmNG-WT CA labeled G89V virions or after SAP (+SAP) and CsA treatment shows insufficient release of CDR from cores. Data in (C and F) is shown as mean and SEM. Stats, Student *t*-test (C) and non-parametric Mann-Whitney Rank sum test (F).
(TIFF)

**S5 Fig. Estimating the fraction of vRNP-packaged CDR in HIV-1 virions.** Chimeric HIV-1 pseudovirus with a mixed composition of K203A/G89V capsid were produced by co-transfecting 293T cells with indicated proportions of pR9ΔEnv plasmids encoding for K203A or G89V CA, along with constructs expressing fluorescent INmNG and CDR labels. Single virus particles were bound on poly-l-lysine cover glass and the incorporated CDR-fluorescence was quantitated. **(A)** Schematics of CDR incorporation into G89V-virions containing an increased proportion of K203A mutant CA, and the release of CDR/CA upon saponin and CsA treatment. **(B)** Images and **(C)** fluorescence intensity analysis of CDR incorporation into G89V/K203A mixed CA containing single virions (top panels in B, and black lines in (C)), in cores after saponin treatment (+SAP; middle panel in (B), and red lines in (C)) and in vRNPs after CsA treatment (+CsA; bottom panel in (B), and blue lines in (C)). Images show CDR alone (left panels) and merged with INmNG (green, right panels). Data in (C) is median at 95% confidence interval of sum CDR fluorescence from >1200 single virus particles. Scale bar in (B) is 2 μm. **(D)** The fraction of CDR fluorescence retained in chimeric cores (+SAP) and in vRNPs (+SAP+CsA) was estimated by normalizing to CDR-fluorescence in 100% K203A/CA virions (set to 100). Normalized values are shown in inset.
(TIFF)

**S6 Fig. HIV-1 CA co-precipitates with vRNP proteins.** HIV-1 virion lysates were immuno-precipitated using antibodies against vRNP components IN, RT, and NC proteins. Anti-CA and anti-gp120 envelope antibodies were used as controls. Mock immuno-precipitation was done without virus lysate to control for false-positives due to antibody-antibody interactions. The immuno-blot was probed with HIV-IgG human serum (bottom) to identify HIV-1 proteins based on their respective molecular weight. A full immuno-blot relative to the cropped images in Fig 4 is shown. Note the non-specific bands detected in antibody alone control (lanes 1–7) are also detected in the viral lysate IP and migrate differently from distinct RT, IN, NC, CA, and MA proteins that are enriched in lanes with respective antibodies.
(TIFF)

**S7 Fig. HIV-1 VRC-associated CDR signals disappear upon nuclear import.** HIV-1 pseudoviruses were co-labeled with CDR (red) along with one of the vRNP associated CA (CAeGFP,

HGFP-CA or IN-CA (green)) markers and the retention of eGFP and CDR signals in the nucleus of TZM-bl cells was analyzed. **(A)** Schematics showing the following scenarios: (1) displacement of CDR from the lattice and entry of intact cores into the nucleus, in this case both eGFP and the vRNP-associated CDR signals should remain colocalized, and (2) Capsid remodeling/ uncoating should result in the loss of high-avidity CDR bound on the lattice and low-affinity CDR bound to eGFP-tagged CA in vRNPs. **(B)** Single z-slice images showing the detection of HIV-1 labeled with eGFP-tagged CA or INmNG-CA and CDR in the cytoplasm and nucleus of TZM-bl cells at 4 h post infection (MOI 0.2). White dashed lines show CA or IN puncta in the nucleus that does not contain CDR signals and yellow arrow heads point to CDR colocalized with eGFP-tagged CA or IN puncta at the NE and cytoplasm. Scale bars 5 µm. **(C)** Quantification of CDR signals in single cores upon CsA treatment *in vitro* and in nuclear VRCs (nucVRCs) at 4 hpi in TZM-bl cells. The retained CDR signal in cores after CsA treatment was used as a comparison for vRNP-associated CDR signals in virions. Data in (C) is median and SEM from 3 independent experiments. Statistics in (C), Mann-Whitney rank sum test.
(TIFF)

**S8 Fig. HIV-1 nuclear import and integration targeting is associated with drastic changes to the structure and composition of the viral core. (A)** The average number of IN-labeled VRCs inside the nucleus and at the nuclear envelope (NE) was determined in TZM-bl cells at 6 hpi (MOI 0.2) as in Fig 6. **(B)** HIV-1 integration targeting into SPADs was determined in 293T cells with HIV-1 viruses containing 100% WT or N74D or a indicated mixture of WT and N74D mutant CA. Genomic DNA was extracted 120 hpi and % SPAD integration targeting was determined. The *in silico* simulated random integration control (RIC) was 2.5% and is shown as dashed blue line. Statistics: Black *vs* 100% WT CA and in Red *vs* 100% N74D CA. Data is mean from 2 independent experiments.
(TIFF)

**S9 Fig. A subset of WT CA in mixed WT/N74D chimeric capsids is sufficient for nuclear speckle localization. (A)** Single Z-plane images showing IN-labeled HIV-1 VRCs (green puncta) colocalized with SC35-nuclear speckles (red) and nuclear envelope (NE)-proximal VRCs in fixed TZM-bl cells at 4 hpi (MOI 2). Note, SC35-nuclear speckles (red) were detected by immunostaining and show borderline overlap with nuclear lamin (blue) Cy5 fluorescence. Scale bar is 5 µm. **(B)** Quantification of nuclear WT/N74D CA VRCs detected inside >0.5 µm of the NE and inside SC35+ nuclear speckles. The ratio of psPAX2-WT CA: pHIV-N74D CA backbone plasmids in 293T transfections are shown above panels in (A), and the respective proportions are shown in x-axis of (B). Note the 100 and 0% WT/CA in (B) corresponds to viruses produced respectively with pHIV-WT CA and pHIV-N74D CA backbones. Data is mean and SEM from 1 of 2 independent experiments, n>120 nuclei analyzed for each condition.
(TIFF)

**S10 Fig. N74D CA mutant becomes increasingly sensitized to restriction by nuclear pore associated IBD-fusion proteins.** Fold-restriction to single round infectivity of HIV-1 virus with mixed N74D and WT CA proteins suggests an increased restriction of N74D CA containing viruses that remain at the NE. Infectivity of VSV-G pseudotyped HIV-1 containing 100% WT or N74D CA and indicated mixture of N74D/WT CA were analyzed at 72 h in MT4-cells stably expressing NETC-GFP-IBD WT or IBD-D366N mutant protein. Fold-restriction was calculated by dividing the luciferase RLU from IBD-D366N expressing cells from IBD-WT

restriction competent cells. Error bars are standard deviation, from 3 experiments.
(TIFF)

**S1 Table. SPAD-targeted integration by phenotypically mixed WT/N74D CA viruses.**
HIV-1 integration sites were determined in 293T cells infected with viruses containing the
indicated proportions of WT CA incorporated into N74D CA viruses. 0% WT/CA indicates a
virus with 100% N74D/CA. Statistical analysis, P values were calculated by Fisher's exact test
in the indicated pairwise manner. The random integration control (RIC) was calculated in sil-
ico. The % of integration sites in SPADs for each sample is shown to the right. Note: P values
from the row corresponding to 100% WT/CA and the column corresponding to 0% WT/CA
were used in S8B Fig to show statistical significance vs WT (black) and vs N74D (red) viral
integration sites, respectively.
(XLSX)

**S2 Table. HIV-1 N74D/CA integration targeting into SPADs by vRNP-associated
INmNG-CA fusion proteins.** Wild-type (WT) or N74D CA mutant HIV-1 was labeled with
Vpr-INmNG or Vpr-INmNG-CA, and the ability of the viral constructs to target SPADs for
integration was determined in HEK293T cells at 5 days post infection (MOI 1). As controls,
unlabeled virus of respective WT and N74D/CA backbones (noLabel) was used. The total
number of unique integration sites, the number of integrations in SPADs, and the % of SPAD
targeted integration for each sample is shown. The random integration control (RIC) was cal-
culated *in silico*. The data shows that vRNP-associated INmNG-CA is unable to stimulate
CPSF6-mediated SPAD-proximal integration of N74D virus.
(XLSX)

## Acknowledgments

We gratefully acknowledge Jeremy Luban (University of Massachusetts Medical School),
Vineet KewalRamani (NCI), Vinay Pathak (NCI), Thomas Hope (Northwestern University)
and the NIH AIDS Reagent Program for reagents. We are grateful to Mariana Marin (Emory
University), and Joao Mamede (Rush University) for critical reading of the paper and helpful
comments. We also thank members of the Cereseto Lab (University of Trento, Italy): Antonio
Casini, Tiziana Coradin, Manuel Nicolussi, and Stephen Findlay Wilson; Melikyan Lab
(Emory University): Hui Wu, Satya Prakash Singh and Mathew Prellberg for technical
assistance.

## Author Contributions

**Conceptualization:** Ashwanth C. Francis, Anna Cereseto, Gregory B. Melikyan.

**Data curation:** Ashwanth C. Francis, Parmit K. Singh, Gregory B. Melikyan.

**Formal analysis:** Ashwanth C. Francis, Jiong Shi, Alan N. Engelman, Christopher Aiken.

**Funding acquisition:** Ashwanth C. Francis, Alan N. Engelman, Christopher Aiken, Gregory
B. Melikyan.

**Investigation:** Ashwanth C. Francis, Anna Cereseto, Parmit K. Singh, Jiong Shi, Alan N.
Engelman, Christopher Aiken, Gregory B. Melikyan.

**Methodology:** Ashwanth C. Francis, Anna Cereseto, Eric Poeschla, Alan N. Engelman, Chris-
topher Aiken.

**Project administration:** Ashwanth C. Francis.

**Resources:** Ashwanth C. Francis.

**Supervision:** Ashwanth C. Francis, Anna Cereseto, Alan N. Engelman, Christopher Aiken, Gregory B. Melikyan.

**Validation:** Ashwanth C. Francis, Gregory B. Melikyan.

**Visualization:** Ashwanth C. Francis, Parmit K. Singh, Gregory B. Melikyan.

**Writing – original draft:** Ashwanth C. Francis, Gregory B. Melikyan.

**Writing – review & editing:** Ashwanth C. Francis, Alan N. Engelman, Christopher Aiken, Gregory B. Melikyan.

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
