## [Decision Letter · Decision Letter 0]

14 Apr 2022

Dear Dr. Francis,

Thank you very much for submitting your manuscript "Localization and functions of capsid and eGFP-tagged capsid proteins in HIV-1 particles" for consideration at PLOS Pathogens. As with all papers reviewed by the journal, your manuscript was reviewed by members of the editorial board and by 3 independent reviewers. In light of the reviews (below this email), we would like to invite the resubmission of a significantly-revised version that takes into account the reviewers' comments.

All 3 reviewers felt that the results were interesting and provocative. They all however had some concerns about the conclusions drawn from the data and suggested a number of controls. In addition, the reviewers felt that the manuscript would be of higher significance if the role that vRNP-associated CA might be playing in infection was determined.

We cannot make any decision about publication until we have seen the revised manuscript and your response to the reviewers' comments. Your revised manuscript is also likely to be sent to reviewers for further evaluation.

Sincerely,

Susan R. Ross, PhD

Section Editor

PLOS Pathogens

Susan Ross

Section Editor

PLOS Pathogens

Kasturi Haldar

Editor-in-Chief

PLOS Pathogens

orcid.org/0000-0001-5065-158X

Michael Malim

Editor-in-Chief

PLOS Pathogens

orcid.org/0000-0002-7699-2064

Reviewer's Responses to Questions

**Part I - Summary**

Reviewer #1: Dr. Francis and colleagues performed an elegant study on eGFP-capsid labeled HIV particles. Ref 7 states that eGFP labeled capsid is incorporated in the virus cone and using these viruses a theory was proposed that HIV uncoating of intact virus particles occurs in the nucleus: Ref 30 however, using a slightly different labeling approach, concluded that eGFP-CA mainly resides within the viral cone and import of these proteins in the nucleus is not indicative of nuclear import of intact cones.

In this paper Francis compared the two theories. In a set of experiments they prove that the interpretation of Ref 30 is correct; eGFP-CA resides within the cone and retaining this fraction in the nucleus does not imply uncoating in the nucleus. A stronger conclusion about this finding is even warranted. The present paper refutes major evidence for a nuclear uncoating event as expressed in Ref 7.

Comments

1. Abstract; eGFP-CA is not a relevant marker for uncoating; replace by is not a “direct” marker of uncoating, it may be used as an indirect marker (some loss during nuclear import, altered accessibility)

2. Intro: the statement that HIV intranuclear transport is determined by CA-CSPF6 interaction should be rephrased taking the results of this paper into account. Does intranuclear CA-CPSF6 interaction relate to cone capsid or within cone capsid? Is there any evidence that capsid-IN complexes in the nucleus are functional?

3. Figure 1 A, B add also plasmid eGF P-CA labeling scheme of Ref 30

4. Figure 1 G what is the eGFP fluorescence in cytoplasmatic PICs? What is the colok IN/Capsid in nucleus vs cytoplasma

5. P4 Zurnic et al (Ref 30) already concluded that labeled capsid was present within the cone; this manuscript is conforming this result; the finding is a confirmation not an entirely novel finding; this should be added to the intro.

6. Unfortunately, few studies were done on the functionality of nuclear complexes containing eGFP-CA. The loss of fluorescence intensity together with IN is taking place at late time points 12 hr, well beyond expected time of integration. When do IN complexes without capsid disappear? What is the location of IN complexes without eGFP-CA? What happens if integration is blocked by integrase inhibitors? This control experiment should be done.

7. Can a 1% eGFP-mutant CA contribution to the cones be excluded in the restriction assays?

Reviewer #2: When, where and how the HIV core uncoats is highly controversial and has been difficult to study due to the lack of direct methods available. In this manuscript, Francis et al. investigate the incorporation of an eGFP-tagged CA construct recently utilised to research uncoating. Rather than being incorporated into the capsid shell of HIV, they find that eGFP-CA is excluded from the shell and instead associates with the vRNP. This suggests that eGFP-CA is not a good tool to study uncoating and questions the interpretation of experimental data acquired using this construct. This is an important finding to alert the field to and is likely to promote much further discussion. In addition, Francis et al. use various mutants to further address uncoating and agree with other recent papers that capsid remodelling is the predominant pathway for nuclear entry.

In general, this topic is of great interest in the field and is likely to be relevant to many researchers. The introduction is well written and gives a good account of the current ideas. Unfortunately, the results section is dense and hard to read partly because of the various construct names and partly due to the complex experimental set ups. The authors have obviously tried to address this by including explanatory diagrams in the figures but perhaps it would help to include a few more sentences summarising what the previous section means in more general/overview terms and what the potential outcomes of the next set of experiments could show (ie how the experiment will address different possible scenarios)? Many of the experiments rely on assumptions about what individual techniques are measuring, so although the authors debunk one tool for studying uncoating, they reply on other tools that also have caveats. This is a general limitation of the field but the authors should take care to list and consider alternative explanations for their data, particularly in the discussion section.

Annoyingly, there are no line numbers!

Reviewer #3: This is an interesting study by a collaborative group, which looks into the distribution of the HIV-1 CA protein in the virion. CA is canonically known to make up the capsid shell that surrounds and protects the viral core vRNP particle, which contains the genome and associated replication enzymes. The authors report evidence that some CA is associated with the vRNPs; this is something that has been alluded to before, but seems established here with sufficient rigor.

To this reviewer, the most significant finding here is that some proportion of CA is associated with the vRNPs, and enters the nucleus as “cargo” rather than as part of the capsid. Unfortunately, this finding is incremental, because it has been suggested before and the authors do not go significantly beyond the phenomenology to elucidate what the vRNP-associated CA is doing. The final set of experiments showing that nuclear transport is sensitive to changes in CA composition is intriguing - WT CA doped into N74D capsid is shown to confer restriction to CPSF6-358 at 10% level, but cannot rescue nuclear import at 50% level - but of unclear significance, as no follow-up is done to determine whether the effect is mediated by capsid-associated or vRNP-associated CA.

The second aspect of the paper is the claim that eGFP-labeled CA is excluded from the capsid. This is a distinct question (eGFP-CA can be associated with the vRNP without necessarily being excluded from the capsid). The data presented do not rigorously establish this. While somewhat of academic interest, this is nevertheless an important issue because eGFP labeling is now being used as a reporter of uncoating.

The paper will be significantly improved by having a clear mechanistic focus - either on vRNP-associated CA or on capsid-associated CA; as it stands, I think the impact is quite modest.

**Part II – Major Issues: Key Experiments Required for Acceptance**

Reviewer #1: Unfortunately, few studies were done in this manuscript on the functionality of nuclear complexes containing eGFP-CA. The loss of fluorescence intensity together with IN is taking place at late time points 12 hr, well beyond expected time of integration. When do IN complexes without capsid disappear? What is the location of IN complexes without eGFP-CA? What happens if integration is blocked by integrase inhibitors? This control experiment should be done.

Reviewer #2: 1) Figure 3: This experiment should be repeated with the G89V mutant used in Supp Figure 4 in combination with the K203A mutation so that only the trapped CA is labelled with CDR. This would give a better indication of the amount of CA inside the core and when it is released, by SAP and CsA, which is an integral part of this paper.

2) Related to point1, Supp Figure 6: Why does the GFP signal remain when the CDR signal is lost upon nuclear entry? I believe there is evidence that CypA binds monomeric CA tightly, so the authors explanation of “strong binding to cores and weak binding to monomeric CA” is not correct. Therefore, in the absence of CsA, CypA should remain bound to CA. Perhaps combining this result with Fig3 suggests that the CDR does not actually bind the GFP-tagged CA at all, but some CDR just gets trapped into cores passively. This is then released when cores uncoat? I think the authors make a few too many assumptions here about how their CDR works and so miss other potential explanations?

3) Figure 4: I’m not sure of the validity of the conclusions from the IP experiments done here. Only a fraction of core proteins are pulled down which could be incompletely lysed particles. It seems more likely that all the core proteins come down because the authors are pulling down at least partially intact cores? By what mechanism do the authors suggest that IN could IP RT and vice versa, as they bind different nucleic acids? Importantly, these particles have not reverse transcribed yet (I believe they are just viral particles in vitro), so there is no vDNA? This is the only evidence the authors can provide to explain how CA could stay associated with IN after uncoating, but it is extremely weak.

Reviewer #3: (1) The authors report a provocative result in Fig 2, in which co-incorporated CA (10%) is tested for its ability to render a capsid made of 90% resistant CA (N74D) susceptible to CPSF6 restriction. A similar experiment is done in Suppl Fig 2, with CypA. The authors conclude, from the differences in behavior of the WT CA-doped vs tagged CA-doped cores that eGFP-labeled CA is excluded from the capsid lattice. This is a critical experiment, but the authors do not show the minimum amount WT CA needs to be incorporated into the mutant capsid for it to be susceptible to restriction. That number estimates the upper limit of incorporation of eGFP-labeled CA into the capsid that would give a false negative in their assays.

(2) Page 9: Experiments reported here can be explained by a simple model in which passage through the nuclear pore channel strips the bound CDR from the capsid surface, even if the capsid remains fully intact. The claim that the capsid undergoes remodeling or is permeabilized during import may be true, but is not rigorously established by the presented data. More evidence is needed.

**Part III – Minor Issues: Editorial and Data Presentation Modifications**

Reviewer #1: 1. Abstract; eGFP-CA is not a relevant marker for uncoating; replace by is not a “direct” marker of uncoating, it may be used as an indirect marker (some loss during nuclear import, altered accessibility)

2. Intro: the statement that HIV intranuclear transport is determined by CA-CSPF6 interaction should be rephrased taking the results of this paper into account. Does intranuclear CA-CPSF6 interaction relate to cone capsid or within cone capsid? Is there any evidence that capsid-IN complexes in the nucleus are functional?

3. Figure 1 A, B add also plasmid eGFP-CA labeling scheme of Ref 30

4. Figure 1 G what is the eGFP fluorescence in cytoplasmatic PICs? What is the coloc IN/Capsid in nucleus vs cytoplasma?

5. P4 Zurnic et al (Ref 30) already concluded that labeled capsid was present within the cone; this manuscript is conforming this result; the finding is a confirmation not an entirely novel finding; this should be added to the intro.

6. Can a 1% eGFP-mutant CA contribution to the cones be excluded in the restriction assays?

Reviewer #2: 1) Page 4, last paragraph: The sentence “Since MA-CAeGFp lacks NC and P6, it is incorporated into particles alongside unlabelled CA/Gag” is confusing. Do the authors mean tagged CA is incorporated “by binding unlabelled CA”? More explanation on how this protein is incorporated into both virions and mature cores should be provided.

2) Figure 1: In previous reports (for example studies by the Hope lab) ~50% of Vpr-GFP is lost from particles following membrane fusion. Why is more of the Vpr-IN retained here? Are the authors proposing the Vpr-IN fusion associates with wt IN after maturation so that it is preferentially retained inside the core? As this is the authors control for particles, it would be good to add more explanation of how this marker is behaving.

3) Figure 1: If the MOI is 0.5, why are there so many dots following the IN or CA staining? Some explanation of what the authors think the dots are is needed.

4) Figure 6B-C: These experiments should be done using the CAeGFP construct to determine whether CAeGFP colocalises with the VRC in nuclear speckles

5) Supp Figure 6: Figure legend says 6hpi but text says 4hpi.

6) I think more deliberation of the experiments in Fig 3, 4 and Supp Fig 6 is needed in the discussion section

Reviewer #3: (1) Page 4 – par 2, line 4-5: “did not noticeably affect overall Gag/GagPol cleavage patterns” – this statement is not supported by Suppl. Fig 1, which shows that CA staining is more intense than Gag. Fid 1D,E should include this important control, and the data should be described accurately and interpreted accordingly.

(2) Fig 6A: The authors should test percentages between 10% and 0%.

(3) Page 4 – par 1, line 5: “N-terminally labeled HGFP-CA and C-terminally labeled CAeGFP”

(4) Page 4 – par 2: Please explain what is being visualized in Fig 1c – it seems these are purified virions, but the text should just say this clearly

(5) Fig 1F – “normalized to unlabeled HIV-1” – please explain in the text or methods how normalization was done

(6) Page 6 – CPSF6, SEC24C and NUP153 do not interact with “the binding pocket between neighboring hexamers”; the pocket is intra-hexameric

(7) Please include line numbers to facilitate reviewing and commenting

PLOS authors have the option to publish the peer review history of their article (what does this mean?). If published, this will include your full peer review and any attached files.

Reviewer #1: No

Reviewer #2: No

Reviewer #3: No
---

## [Decision Letter · Decision Letter 1]

21 Jul 2022

Dear Dr. Francis,

We are pleased to inform you that your manuscript 'Localization and functions of native and eGFP-tagged capsid proteins in HIV-1 particles' has been provisionally accepted for publication in PLOS Pathogens.

Best regards,

Susan R. Ross, PhD

Section Editor

PLOS Pathogens

Susan Ross

Section Editor

PLOS Pathogens

Kasturi Haldar

Editor-in-Chief

PLOS Pathogens

orcid.org/0000-0001-5065-158X

Michael Malim

Editor-in-Chief

PLOS Pathogens

orcid.org/0000-0002-7699-2064

Reviewer Comments (if any, and for reference):

Reviewer's Responses to Questions

**Part I - Summary**

Reviewer #1: Strength: paper shows that eGFP-CA is present within the capsid cone and cannot be used as a direct marker of HIV uncoating; this implies that previous findings using this labelling method to pinpoint uncoating to the nucleus are probably wrong

Reviewer #2: I am happy with the modifications the authors have made to the manuscript and think they improved the paper.

Reviewer #3: The authors have addressed my most substantive concerns.

**Part II – Major Issues: Key Experiments Required for Acceptance**

Reviewer #1: Comments have been addressed

Reviewer #2: None

Reviewer #3: (No Response)

**Part III – Minor Issues: Editorial and Data Presentation Modifications**

Reviewer #1: Comments have been addressed

Reviewer #2: None

Reviewer #3: (No Response)

PLOS authors have the option to publish the peer review history of their article (what does this mean?). If published, this will include your full peer review and any attached files.

Reviewer #1: No

Reviewer #2: No

Reviewer #3: No

---

## [Editor Report · Acceptance letter]

7 Aug 2022

Dear Dr. Francis,

We are delighted to inform you that your manuscript, "Localization and functions of native and eGFP-tagged capsid proteins in HIV-1 particles," has been formally accepted for publication in PLOS Pathogens.

Best regards,

Kasturi Haldar

Editor-in-Chief

PLOS Pathogens

orcid.org/0000-0001-5065-158X

Michael Malim

Editor-in-Chief

PLOS Pathogens

orcid.org/0000-0002-7699-2064